# Sensory restoration by epidural stimulation of the lateral spinal cord in upper-limb amputees

**Santosh Chandrasekaran**[1,2,3†]**, Ameya C Nanivadekar**[1,3,4†]**, Gina McKernan**[2,5]**, Eric R Helm**[2]**, Michael L Boninger**[1,2,4,5,6]**, Jennifer L Collinger**[1,2,3,4,5]**, Robert A Gaunt**[1,2,3,4]**, Lee E Fisher**[1,2,3,4]*****

[1]Rehab Neural Engineering Labs, University of Pittsburgh, Pittsburgh, United States; [2]Department of Physical Medicine and Rehabilitation, University of Pittsburgh, Pittsburgh, United States; [3]Center for Neural Basis of Cognition, Pittsburgh, United States; [4]Department of Bioengineering, University of Pittsburgh, Pittsburgh, United States; [5]Human Engineering Research Labs, VA Center of Excellence, Department of Veteran Affairs, Pittsburgh, United States; [6]University of Pittsburgh Clinical Translational Science Institute, Pittsburgh, United States

**Abstract** Restoring somatosensory feedback to people with limb amputations is crucial to improve prosthetic control. Multiple studies have demonstrated that peripheral nerve stimulation and targeted reinnervation can provide somatotopically relevant sensory feedback. While effective, the surgical procedures required for these techniques remain a major barrier to translatability. Here, we demonstrate in four people with upper-limb amputation that epidural spinal cord stimulation (SCS), a common clinical technique to treat pain, evoked somatosensory percepts that were perceived as emanating from the missing arm and hand. Over up to 29 days, stimulation evoked sensory percepts in consistent locations in the missing hand regardless of time since amputation or level of amputation. Evoked sensations were occasionally described as naturalistic (e.g. touch or pressure), but were often paresthesias. Increasing stimulus amplitude increased the perceived intensity linearly, without increasing area of the sensations. These results demonstrate the potential of SCS as a tool to restore somatosensation after amputations.

**\*For correspondence:**
lef44@pitt.edu

†These authors contributed equally to this work

**Competing interests:** The authors declare that no competing interests exist.

## Introduction

Individuals with amputations consistently state that the lack of somatosensory feedback from their prosthetic device is a significant problem that limits its utility (*Cordella et al., 2016*) and is often a primary cause of prosthesis abandonment (*Biddiss and Chau, 2007*; *Wijk and Carlsson, 2015*). In the case of upper-limb amputations, the absence of somatosensory feedback particularly affects the ability to generate the finely controlled movements that are required for object manipulation (*Cordella et al., 2016*; *Lundborg et al., 1998*; *Pylatiuk et al., 2007*; *Wijk and Carlsson, 2015*). Although sophisticated myoelectric prostheses with multiple degrees of freedom (*Belter et al., 2013*) are becoming increasingly available, their potential is limited because they provide little or no somatosensory feedback (*Biddiss et al., 2007*; *Biddiss and Chau, 2007*; *FAAOP et al., 2015*; *Peerdeman et al., 2011*). In fact, body-powered devices are often preferred because of the feedback they provide through their harness and cable system (*Huang et al., 2001*; *Stark and LeBlanc, 2004*; *Uellendahl, 2000*; *Williams, 2011*). Partially addressing this limitation, advanced robotic prosthetic arms have been designed with embedded sensors that could be harnessed to provide somatosensory signals to a neural interface (*Cipriani et al., 2011*; *Perry et al., 2018*; *Saudabayev and Varol, 2015*). Thus, developing a robust and intuitive means to deliver

**eLife digest** Even some of the most advanced prosthetic arms lack an important feature: the ability to relay information about touch or pressure to the wearer. In fact, many people prefer to use simpler prostheses whose cables and harnesses pass on information about tension. However, recent studies suggest that electrical stimulation might give prosthesis users more sensation and better control.

After an amputation, the nerves that used to deliver sensory information from the hand still exist above the injury. Stimulating these nerves can help to recreate sensations in the missing limb and improve the control of the prosthesis. Still, this stimulation requires complicated surgical interventions to implant electrodes in or around the nerves. Spinal cord stimulation – a technique where a small electrical device is inserted near the spinal cord to stimulate nerves – may be an easier alternative. This approach only requires a simple outpatient procedure, and it is routinely used to treat chronic pain conditions.

Now, Chandrasekaran, Nanivadekar et al. show that spinal cord stimulation can produce the feeling of sensations in a person's missing hand or arm. In the experiments, four people who had an arm amputation underwent spinal cord stimulation over 29 days. During the stimulation, the participants reported feeling electrical buzzing, vibration, or pressure in their missing limb. Changing the strength of the electric signals delivered to the spinal cord altered the intensity of these sensations.

The experiments are a step toward developing better prosthetics that restore some sensation. Further studies are now needed to determine whether spinal cord stimulation would allow people to perform sensory tasks with a prosthetic, for example handling an object that they cannot see.

somatosensory information to the nervous system is an important endeavor to ensure the adoption and use of the latest advancements in prosthetics.

Several research groups have explored the potential of peripheral nerve stimulation to provide sensory feedback to people with amputation and examined the effects of feedback on prosthetic control. Sensory restoration has been achieved using a variety of neural interfaces including epineural cuff electrodes like the spiral cuff (*Ortiz-Catalan et al., 2020*; *Ortiz-Catalan et al., 2014*) and flat interface nerve electrode (*Tan et al., 2014*) or microelectrodes that penetrate the epineurium, such as the longitudinal intrafascicular electrode (*Horch et al., 2011*), transverse intrafascicular multichannel electrode (*Raspopovic et al., 2014*), or Utah slant array (*Davis et al., 2016*). Targeted sensory reinnervation is another approach that can allow vibrotactile or electrotactile feedback on the residual limb to be perceived as emanating from the missing limb (*Marasco et al., 2011*; *Marasco et al., 2009*). This is achieved by first surgically redirecting nerves that formerly innervated the missing limb to patches of skin on the residual limb or elsewhere, and then providing electrical or mechanical stimulation to the newly innervated site (*Kuiken et al., 2007a*; *Kuiken et al., 2007b*). These approaches can evoke focal sensations that are perceived to emanate from the upper-limb, even decades after injury, and can improve the control of prosthetic limbs. However, all of these approaches involve specialized electrodes and/or surgeries that are not part of common surgical practice. Further, these approaches often target nerves in the distal limb, which could limit their use in people with proximal amputations such as shoulder disarticulations.

Spinal cord stimulation (SCS) systems are an FDA-approved, commercially available technology that could potentially be used to restore somatosensation. SCS leads are currently implanted in approximately 50,000 patients every year in the USA to treat chronic back and limb pain (*Kumar and Rizvi, 2014*). The standard clinical approach begins with a week-long trial phase with temporarily implanted leads, and if patients experience pain relief, permanent implantation occurs during an hour-long follow-up procedure. For the trial phase, SCS leads are inserted percutaneously into the epidural space on the dorsal side of the spinal cord via a minimally invasive, outpatient procedure (*Kinfe et al., 2014*). Clinically effective stimulation parameters typically evoke paresthesias (i.e. sensation of electrical buzzing) that are perceived to be co-located with the region of pain. SCS leads are usually placed over the dorsal columns along the midline of the spinal cord which limits the evoked paresthesias to the proximal areas of the trunk and limbs. However, recent studies have

demonstrated that stimulation of lateral structures in the spinal cord and spinal roots can evoke paresthesias that selectively emanate from the distal regions of the body (*Deer et al., 2013*; *Harrison et al., 2018*; *Liem et al., 2013*; *Lynch et al., 2011*), likely by stimulating the same sensory afferent neurons that are targeted by peripheral nerve stimulation for prosthetic applications (*Capogrosso et al., 2013*). As such, these devices provide an attractive option for widespread deployment of a neuroprosthesis than can evoke somatosensory percepts from distal aspects of the amputated limb, including the hand and fingers.

In this study, we implanted percutaneous SCS leads into the lateral epidural space of four people with upper-limb amputations and characterized the sensations evoked when the cervical spinal cord and spinal roots were stimulated. The goals of the study were to demonstrate the feasibility of lateral SCS to restore somatosensation and to guide technical development for future studies that will include full implantation of SCS leads and stimulators. In all subjects, lateral SCS evoked sensations that were perceived to emanate from the missing limb, including focal regions in the hand, regardless of the level of amputation (trans-radial to shoulder disarticulation). These sensations were stable throughout the 29 day testing period and showed only minor changes in area and location. Additionally, in some cases, it was possible to evoke naturalistic, rather than paresthetic sensations, though the incidence of naturalistic sensations varied by subject. Considering these results along with the extensive clinical use of SCS, this approach to somatosensory restoration could be one that is beneficial to a diverse population of amputees, including those with proximal amputations. Further, these percutaneously implanted SCS devices are a useful tool for the development of somatosensory neuroprosthetic systems, especially for research projects that focus on advanced prosthetic control but have not developed their own technologies and techniques for restoring sensory feedback.

## Results

### SCS evokes sensory percepts localized to the missing limb

Three SCS leads were implanted in the cervical epidural space in each of four individuals with upper-limb amputation (*Table 1*). The percutaneous implant was maintained for the full 29 day duration of the study for all subjects except Subject 2, who requested removal of the leads after two weeks due to personal factors and discomfort from caudal migration of one of the leads. We stimulated with both monopolar and multipolar electrode configurations. Stimulus amplitudes, frequencies, and pulse widths ranged 0–6 mA, 1–300 Hz, and 50–1000 µs, respectively.

In all four subjects, epidural SCS evoked sensory percepts in distinct regions of the missing limb including the fingers, palm, and forearm. While some sensory percepts were diffuse and covered the entire missing limb, other percepts were localized to a very specific area, such as the ulnar region of the palm or wrist, or individual fingers. *Figure 1* shows select representative percepts for all subjects; for an interactive visualization of all evoked percepts localized to the missing limb, see *Supplementary file 1*. In Subjects 1 and 2, only multipolar stimulation evoked sensory percepts that were localized to focal regions of the missing hand and fingers (*Figure 1—figure supplement 1*). In Subjects 2 and 3, most percepts were accompanied by a sensation on the residual limb. This was the case even when there was a percept that was focally restricted to a distal region of the missing limb, such as a finger (e.g. purple thumb/shoulder sensation in Subject 2, *Figure 1*). These additional

**Table 1.** Subject information.
Demographic, amputation, and study-related information for each subject.

| Subject | Age | Gender | Amputation characteristics | | | | Implant duration (days) |
|---------|-----|--------|------------|------|-------|-------|-------------------------|
| | | | Years since | Side | Level | Cause | |
| 1 | 67 | Female | >5 | Right | Shoulder disarticulation | Necrotizing fasciitis | 29 |
| 2 | 33 | Male | >16 | Left | Transhumeral | Trauma | 15 |
| 3 | 38 | Female | >2 | Right | Transhumeral | Trauma | 29 |
| 4 | 44 | Female | >3 | Right | Transradial | Compartment syndrome | 29 |

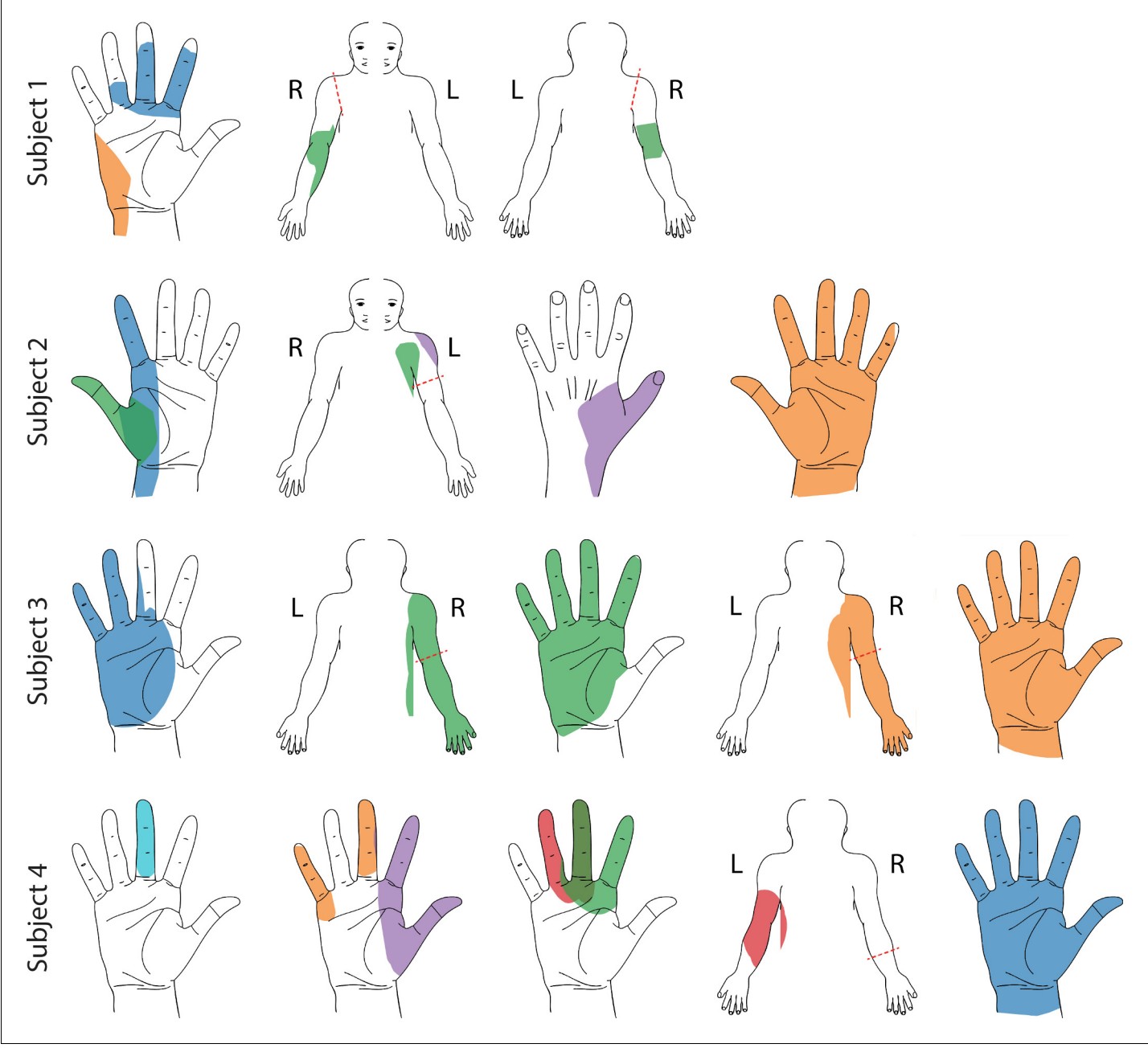

**Figure 1.** Representative sensory percept locations. Colored areas represent selected projected fields that were reported for more than two testing sessions and remained stable for at least two weeks. Each color represents a unique stimulation electrode per subject. If a pair of percepts had more than 70% overlap, only the more focal percept is shown here (*Charkhkar et al., 2018*). For a complete presentation of all sensations included in this study, see interactive *Supplementary file 1*.

The online version of this article includes the following figure supplement(s) for figure 1:

**Figure supplement 1.** Effect of monopolar and multipolar stimulation.

**Figure supplement 2.** Touchscreen interface for describing evoked sensory percepts.

proximal sensations emanated predominantly from the end of the residual limb. The incidence rate of such simultaneous sensations varied from 0% and 8% for Subjects 1 and 4 to 92% and 98% for Subjects 2 and 3. There were also a subset of mono- and multi-polar electrodes that evoked sensations bilaterally or only in the contralateral intact limb (14.3% and 15.4% of all electrodes that generated a sensation across all subjects; n = 447). While these sensations might be useful in a

neuroprosthesis for people with bilateral amputation, they were not a focus of this study and were not included in any of the analyses presented here.

We sought to determine if stimulation of specific regions of the spinal cord consistently evoked sensations that were perceived to emanate from specific regions of the arm and hard across subjects. We hypothesized that the location of the perceived sensation would be driven by the location of the cathodic electrode with respect to the spinal cord according to expected dermatomes. *Figure 2* shows the proportion of sensory percepts in a specific anatomical region (dashed lines, *Figure 2A*) evoked by electrodes situated at each spinal level (*Figure 2B,C*). There were notable similarities between the perceived locations and dermatomes (*Foerster, 1933*; *Lee et al., 2008*), however there was considerable inter-subject variability and sensations were not evoked in all regions of the hand in all subjects (*Figure 2C*). For example, sensations reported in the thumb were predominantly evoked by electrodes located near the C6 root in Subjects 2 and 4 (0%, 67%, 26%, and 50% for Subjects 1–4 respectively). Similarly, a high proportion of the percepts localized to the 2nd and 3rd digits were evoked by electrodes near the C7 root in Subjects 2 and 3 (0%, 50%, 66%,

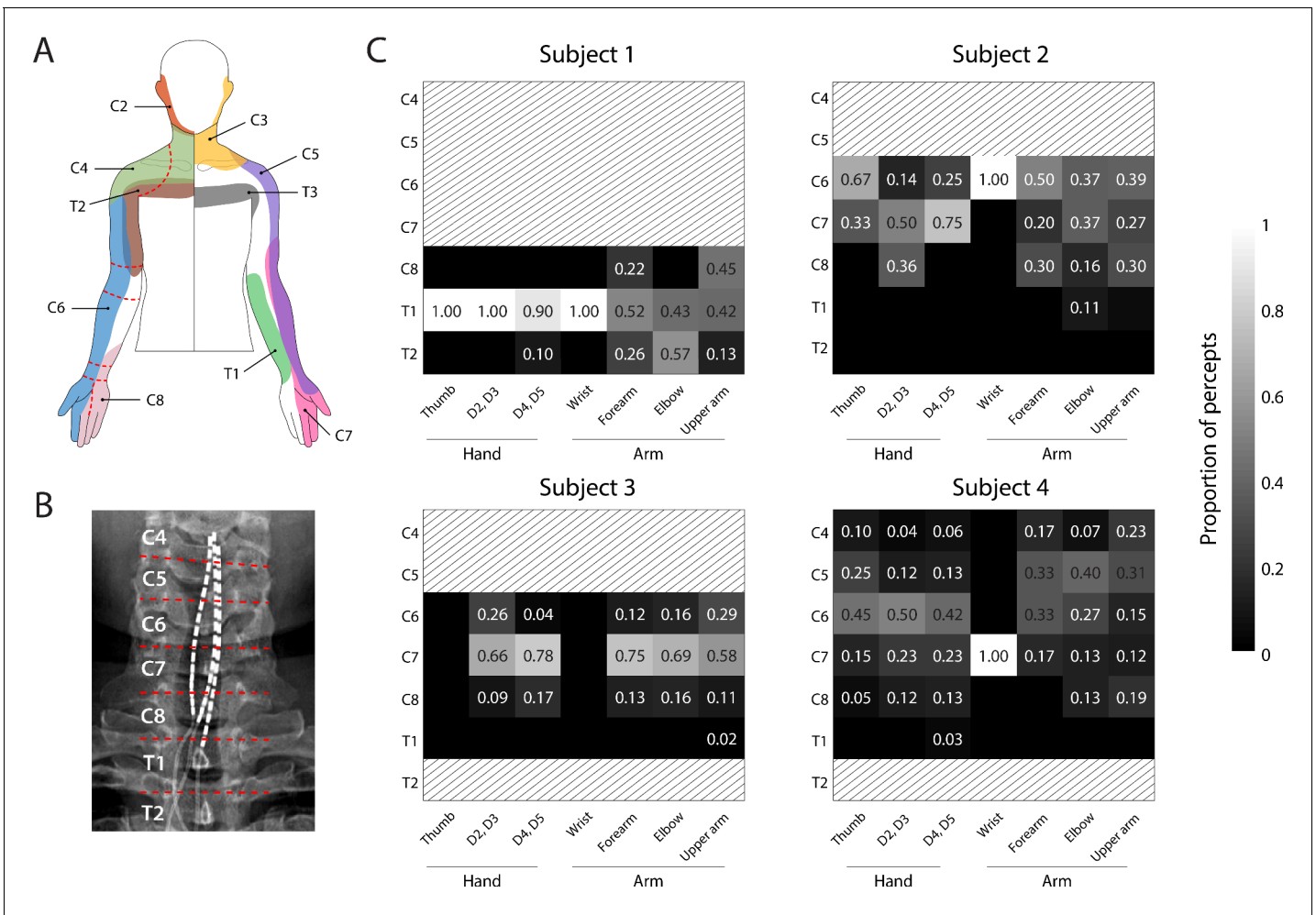

**Figure 2.** Dermatomal organization of the evoked percepts. (A) Schematic of dermatomes, adapted from *Lee et al., 2008*. Overlapping dermatome areas are shown in lighter shades. Dotted lines indicate our division of different regions of the fingers, hand, and arm. (B) An example of the segmentation of the spinal cord (from Subject 4) used to determine the location of each stimulation electrode. (C) Heat maps show the relative proportion of electrodes located at different spinal levels to the total number of percepts emanating from a specific region of the arm. For Subjects 1-3, the spinal level of each electrode was defined by the position of the cathode with respect to the spinal levels as seen in the X-rays. For Subject 4, the spinal level of each electrode was defined by the position of the anode. Spinal levels that have no electrodes nearby are marked with gray hatching.

**Table 2.** Descriptors provided for characterizing the evoked percepts.

The various descriptors that subjects were asked to choose from while describing the modality and intensity of the evoked sensory percept. Visual analog scales (VAS) were presented as a slider bar and no specific numbers were shown.

| Naturalness | Depth | Mechanical | Tingle | Movement | Temperature |
|---|---|---|---|---|---|
| VAS (Totally Unnatural to Totally Natural) | Skin surface | Touch | Electrical | Vibration | VAS (Very Cold to Very Hot) |
| | Below skin | Pressure | Tickle | Body/limb/joint | |
| | Diffuse | Sharp | Itch | Across skin | |
| | Both | | Pins and Needles | | |
| | | VAS (intensity) | VAS (intensity) | VAS (intensity) | |

and 23%, for Subjects 1–4, respectively). However, sensations in 4$^{th}$ and 5$^{th}$ digits (within the C8 dermatome) were evoked predominantly by electrodes near the C7 root in Subjects 2 and 3 (0%, 75%, 78%, and 23% in Subjects 1–4, respectively). Interestingly, for Subject 4, electrodes near the C6 root produced most of the percepts in the hand (2$^{nd}$ and 3$^{rd}$ digits: 52%, 4$^{th}$ and 5$^{th}$ digits: 45%). Moreover, almost all the electrodes in Subject 1, including those that evoked focal percepts in the fingers and palm, were located near the T1 root. Overall, these results demonstrate that, while there was some consistency between the locations of stimulation and dermatomes, there was considerable inter-subject variability in many of the evoked sensations.

We asked the subjects to describe the evoked sensations using a set of words provided in a predefined list (*Table 2*). This allowed us to standardize the descriptions of the percepts across subjects and put them in context of previous research (*Heming et al., 2010*; *Lenz et al., 1993*). Subjects could report more than one modality simultaneously. All sensations that had an 'electrical tingle',' pins and needles', 'sharp', or 'tickle' component were considered paresthetic. If these sensations also included descriptors for mechanical, movement, or temperature modalities, they were considered mixed modality sensations. Sensations that did not include any paresthetic descriptors were considered naturalistic. The unique combinations of percept descriptors used by each subject along with the fraction of naturalistic, paresthetic, and mixed modality sensations are shown in *Figure 3*. For Subjects 1, 2, and 4, most sensory percepts were either paresthetic or of mixed modality (90.2%, 75.2%, and 96.5%, respectively). Subject 1 reported 74.2% of these percepts as purely paresthetic, whereas only a small fraction of these percepts were reported as purely paresthetic by Subjects 2 and 4 (4.1% and 0.3%). For Subject 2 the evoked percept was most frequently described as tingle-pressure and for Subject 4 the evoked percept was most frequently described as tingle-pressure-vibration. Subject 3 predominantly reported naturalistic sensations (79.9%) with most of those percepts described as pure vibration. In fact, for this subject, 80% of all evoked percepts contained a 'vibration' component, and most mixed modality percepts (97.8%) were described as tingle-vibration with only one instance of a purely paresthetic percept.

More naturalistic modalities, like 'touch' and 'pressure', were elicited to varying degrees among the subjects (0.5%, 60.5%, 25.2% and 75.8% of unique stimulation parameter combinations for Subjects 1–4, respectively). Interestingly, sensations described as purely touch or pressure were reported in 8.25% and 19.5% of all evoked percepts in Subjects 2 and 3, respectively. Otherwise, these naturalistic sensations were commonly accompanied by a paresthesia, as particularly seen in Subject 4. Percepts containing a dynamic ('movement') component that may be described as proprioceptive were evoked at least once in all subjects. Subjects were able to describe distinct proprioceptive sensations in the phantom limb such as opening and closing of the hand, movement of the thumb, and flexing of the elbow. However, unlike the tactile percepts that were predominantly stable across days, these proprioceptive sensations could be repeatedly evoked only for a few minutes even with consistent stimulus parameters. Only in the case of Subject 4 (the subject with trans-radial amputation), were we able to evoke sensations of thumb and wrist movement reliably over longer time courses, spanning multiple days and weeks. Interestingly, these proprioceptive percepts were elicited by a set of three closely situated electrodes over a narrow range of stimulus parameters (stimulus amplitude = 3 mA, stimulus frequency = 1–5 Hz).

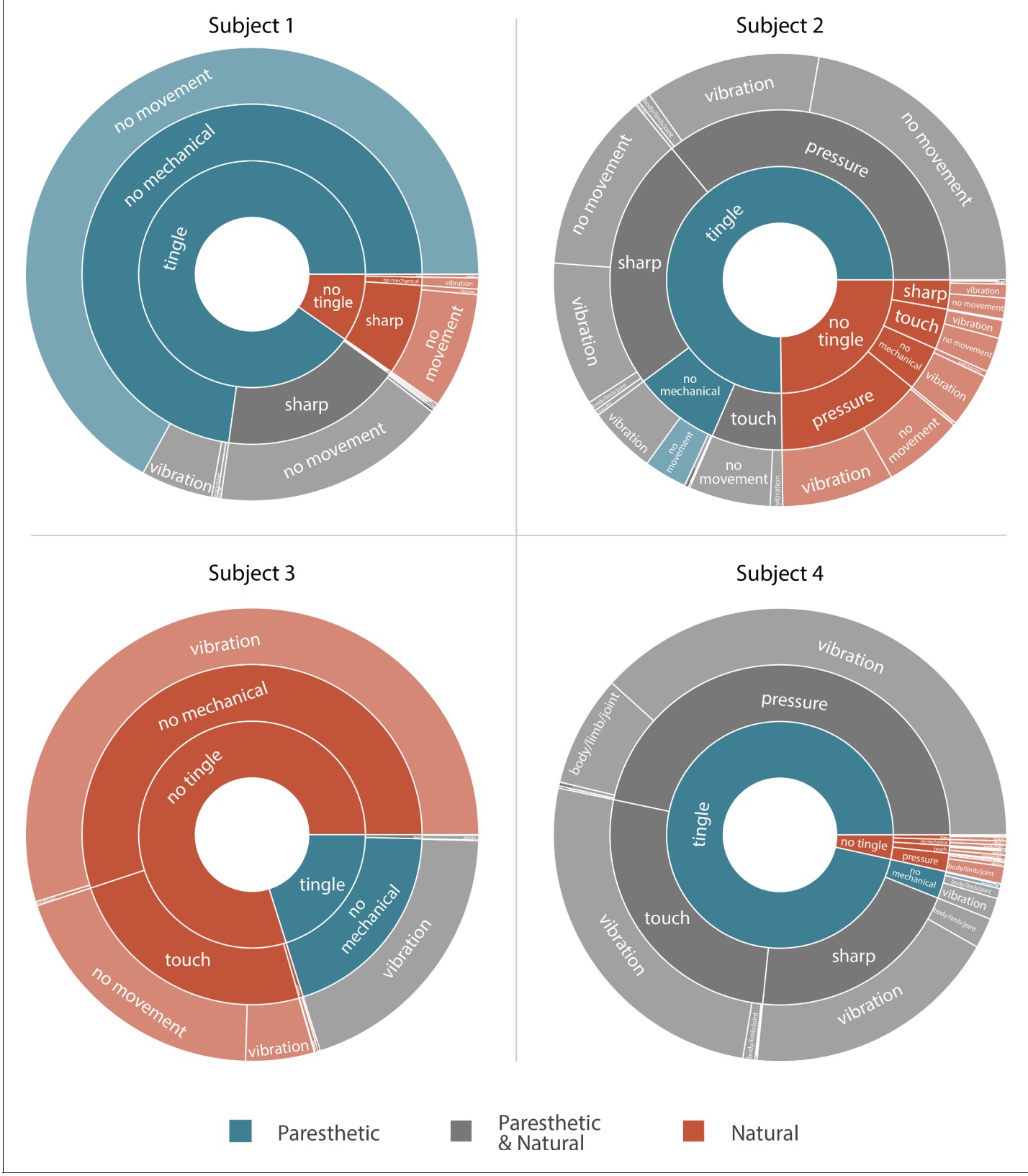

**Figure 3.** Sunburst plot showing the combination of paresthetic (teal), naturalistic (red), and mixed (grey) percept descriptors used by each subject. Each annulus represents a modality descriptor that the subjects could select. The innermost annulus represents sensations where a 'tingle' descriptor was used. The middle annulus represents the fraction of each tingle descriptor that co-occurred with a 'mechanical' descriptor. The outermost annulus

*Figure 3 continued on next page*

*Figure 3 continued*

represents the fraction of tingle and mechanical descriptors that co-occurred with a 'movement' descriptor. For Subjects 1, 2, and 4, most sensory percepts were either paresthetic or of mixed modality (90.2%, 75.2%, and 96.5%, respectively). Subject 3 predominantly reported naturalistic sensations (79.9%) with most of those percepts described as pure vibration. An interactive version of this figure with expandable sectors and annotations for the percent occurrence of each descriptor is available in *Supplementary file 3*. Source data are available in *Figure 3—source data 1*.

The online version of this article includes the following source data for figure 3:

**Source data 1.** Number of times each subject reported each of the different modality descriptor combinations shown in *Figure 3*.

Varying the stimulation frequency influenced the modality of the evoked sensation in Subject 3, but not in the other subjects. For Subject 3, the sensory percepts that were described as 'touch' or 'pressure' occurred in up to 90% of trials at low stimulation frequencies (below 20 Hz) while stimulation frequencies above 50 Hz evoked percepts that were always characterized as mixed modality. Subject 1 never reported these naturalistic sensations, which could be because we never stimulated at frequencies below 20 Hz, while Subjects 2 and 4 reported naturalistic, mixed, and paresthetic sensations independent of the stimulus frequency.

## Psychophysical assessment of evoked percepts

We quantified the detection threshold for sensations in the missing limb in all four subjects using a two-alternative forced-choice paradigm. Because Subjects 2 and 3 frequently experienced co-evoked sensations in the phantom and on the residual limb, for psychophysical assessments, we asked them to focus only on the distal phantom percept whenever stimulation co-evoked a sensation in the residual limb. In this task, the subject reported which of two intervals contained the stimulus train. With a randomized presentation of various stimulation amplitudes, we measured the detection threshold as the minimum amplitude at which the subject could correctly report the interval containing the stimulation train with 75% accuracy (*Figure 4A*). Mean detection thresholds (*Figure 4B*) for Subjects 1–4 were 3.75 mA (n = 2 electrodes), 1.25 ± 0.36 mA (n = 5 electrodes), 1.58 ± 0.39 mA (n = 14 electrodes) and 1.94 ± 0.27 mA (n = 14 electrodes), respectively.

We measured just-noticeable differences (JND) in stimulation amplitude with a two-alternative forced choice task in Subjects 3 and 4. We evaluated the goodness-of-fit using the probability of transformed likelihood ratio (pTLR), which spans 0–1 with a higher value signifying a better fit and values below 0.05 signifying an unacceptable fit. In Subject 3, for one electrode, the subject could perceive a change of 86 µA (slope, $\beta$ = 0.045, pTLR = 0.58) at 75% accuracy when the standard amplitude was 2.5 mA, and a higher standard amplitude of 4 mA increased the JND to 280 µA (slope, $\beta$ = 0.073, pTLR = 0.83; *Figure 4C*). In Subject 4, the JNDs showed a similar dependence on standard amplitude with mean $JND_{2.5}$=60 ± 21 µA (median slope, $\beta$ = 0.040, median pTLR = 0.79) and mean $JND_{4.0}$=338 ± 98 µA (median slope, $\beta$ = 0.005, median pTLR = 0.40, n = 5 electrodes; *Figure 4D* and *Figure 4—figure supplement 1*). To put these numbers in context, for Subject 4, with mean threshold at approximately 2 mA and maximum stimulation amplitude at 6 mA, the JNDs represent 1.3% (at standard amplitude of 2.5 mA) and 9% (at standard amplitude of 4 mA) of the available stimulation range.

To measure the relationship between stimulation amplitude and sensation intensity, subjects performed a free magnitude estimation task, in which they were instructed to rate perceived intensity on an open-ended numerical scale as stimulation amplitude was varied randomly. They were instructed to scale their response such that a doubling in perceived intensity was reported as a doubling in the numerical response. To control for variability across different electrodes and across different testing sessions, we normalized each electrode to the mean of its response. We observed that as stimulation amplitude was increased, the perceived intensity of the sensory percept increased linearly for all subjects; an effect that was consistent across repetitions of the task on multiple days (*Figure 4E*). A linear fit was determined to be better than or at least as good as a sigmoid or logarithmic fit based on adjusted $R^2$ values, and all electrodes had a significant linear relationship between stimulus amplitude and perceived intensity, ($p_{int}$ <0.01, F-test, where $p_{int}$ is the two-sided p-value for the null hypothesis that the slope of the regression line was zero). This linear relationship between amplitude and intensity was maintained even though different electrodes were tested with different pulse widths and frequencies. *Supplementary file 2* shows a complete list of stimulation parameters used for these experiments.

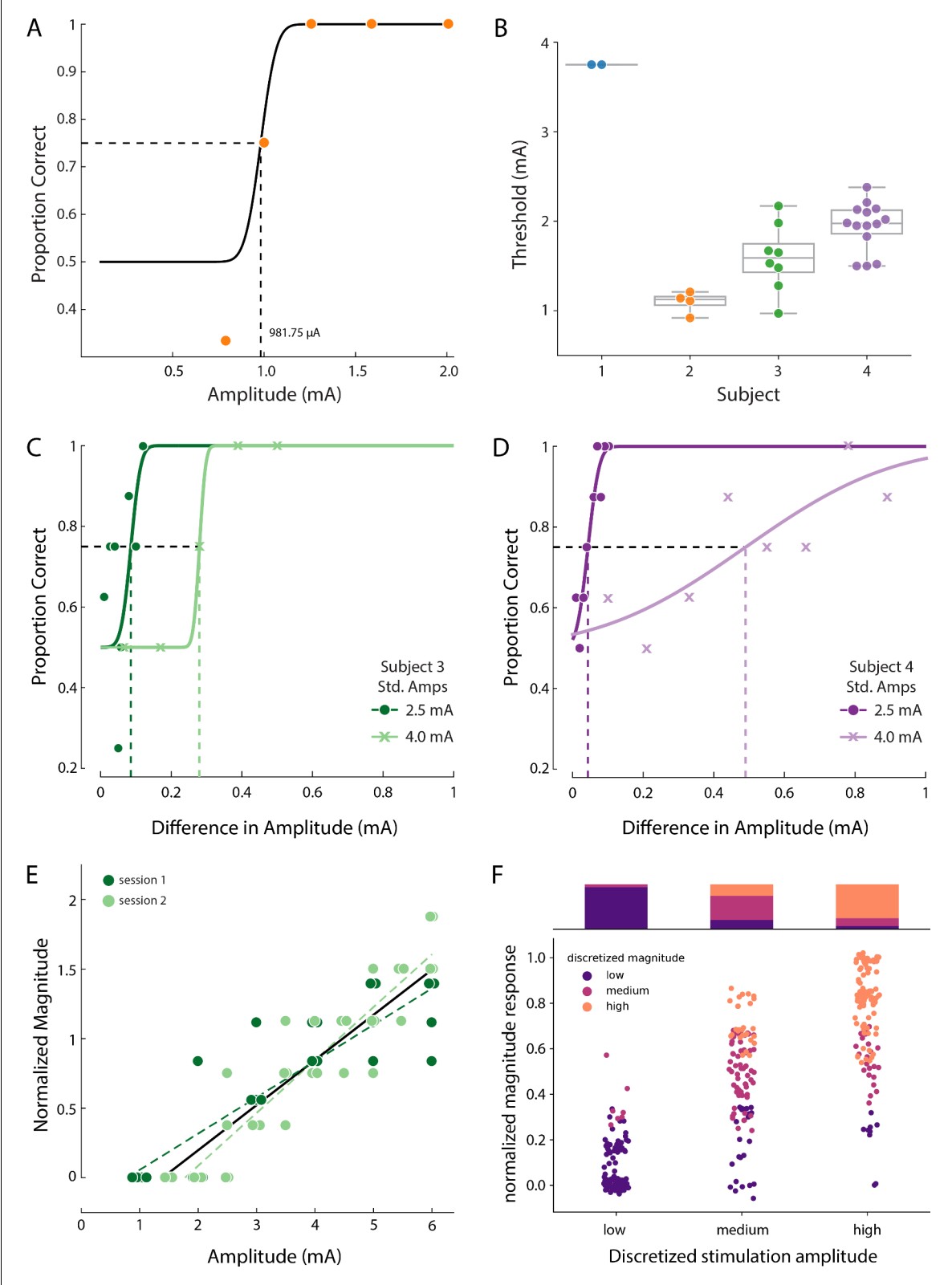

**Figure 4.** Psychophysics of the evoked sensory percepts. (**A**) Example data from a detection task for a single electrode from Subject 2. Data were collected using a threshold tracking method and a psychometric function was fit to the data. The detection threshold was determined to be 982 μA. (**B**) Scatter plot showing the distribution of all the detection thresholds for Subjects 1 (blue), 2 (orange), 3 (green) and 4 (purple). (**C**) Example data for the just-noticeable differences at two different standard amplitudes for one electrode in Subject 3 and (**D**) Subject 4. Error bars represent SD. (**E**) Example

*Figure 4 continued on next page*

*Figure 4 continued*

data from Subject 3 of a free magnitude estimation task carried out on two different days (light and dark green circles) for a single electrode. Data from multiple days were compared after normalizing each electrode to its mean response. Perceived intensity varied linearly with stimulus amplitude for each individual testing session (dashed and solid green lines) as well as when taken together (black solid line). The slope of these lines was measured in units of mA$^{-1}$ (F) Distribution of the stimulation amplitude and the reported intensity of the evoked percept for all subjects. Stimulation amplitude and reported intensity were independently discretized into three linearly spaced (low, medium, and high) bins and subject ratings of sensation intensity accurately predicted these bins. The stacked bar graph represents the percentage of times a binned magnitude response occurred for the corresponding discretized amplitude. Source data for panels B and F are available in *Figure 4—source data 1* and *Figure 4—source data 2* , respectively.

The online version of this article includes the following source data and figure supplement(s) for figure 4:

**Source data 1.** Detection threshold for each electrode included in *Figure 4B*.
**Source data 2.** Magnitude estimation data for each electrode included in *Figure 4E and F*.
**Figure supplement 1.** Psychometric curves fit to response of Subject four to JND tasks on four electrodes as shown in *Figure 4D*.
**Figure supplement 2.** Distribution of the stimulation amplitude and the reported intensity of the evoked percept for all subjects.
**Figure supplement 2—source data 1.** Magnitude estimation data for each electrode included in *Figure 4—figure supplement 2*.

These results taken together show that subjects should be able to perceive graded sensory feedback for linearly spaced gradations greater than the JND for each electrode. The number of gradations in stimulation determines the number of discrete targets (such as identifying three different levels of force) that can be represented in a functional task. To identify the optimal gradation for functionally relevant sensory feedback via SCS, we partitioned stimulation amplitudes and subject responses during the free magnitude estimation task into three or five discrete linearly spaced ranges. The partitioned data were used to estimate how reliably subjects can distinguish sensations for each of these amplitude ranges. *Figure 4F* shows the distribution of the normalized subject responses for a 3-target task where the overall accuracy was 72%. All subjects reported sensations in the low and high range of stimulation with a high accuracy, (79% and 95% accuracy, respectively) with medium targets having an accuracy of 54%. When the data were partitioned into five discrete ranges, the overall accuracy was 46% (*Figure 4—figure supplement 2*). In the context of clinical translation, these results indicate that it may be possible for the user to discriminate three specific intensity levels based on stimulation amplitude alone.

Since we found a consistent linear relationship between percept intensity and stimulation amplitude, we quantified the changes in percept area that occurred as the stimulation amplitude was increased. In a prosthetic device, being able to modulate the percept intensity independent of the area is critical to deliver graded feedback that remains focal. *Figure 5A* shows an example of a percept where the area and centroid remain stable as the stimulation amplitude is increased. To examine the effect of stimulation amplitude on the area and intensity of the evoked percept, we computed the least-squares regression line for area versus stimulation amplitude and obtained the two-sided p-value ($p_{area}$) for the null hypothesis that the slope of the regression line was zero. We also compared the slope of this regression line with the slope of the linear fit between stimulation amplitude and reported intensity (obtained from magnitude estimation trials, *Supplementary file 2*) for each electrode to identify whether percept area and intensity were modulated independently (*Figure 5B*). Across all electrodes (n = 24), the slope of the linear fit for area was less than the slope for intensity. For three electrodes, both area and intensity were modulated by stimulation amplitude ($p_{area}$ <0.01, median β = 0.25 and $p_{int}$ <0.01 median β = 1.36). For the remaining 21 electrodes, only intensity, but not area, was modulated by stimulation amplitude ($p_{area}$ >0.05, median β = 0.01 and $p_{int}$ <0.01, median β = 1.33). This indicates that for most electrodes, it is possible to modulate percept intensity independent of percept area.

## Stability of SCS electrodes and evoked sensory percepts

Lead migration is a common clinical complication for SCS, with an incidence rate as high as 15–20% (*Cameron, 2004*; *Kinfe et al., 2014*; *Mekhail et al., 2011*; *Mironer et al., 2004*). Lead migration would change the location and modality of evoked sensations, which could limit the long-term viability of SCS and would also complicate the scientific utility of percutaneous SCS as a testbed for novel neuroprosthetic techniques. We performed weekly X-rays that allowed us to monitor the position of the leads and quantify migration over the duration of the implant. Superimposing the intraoperative

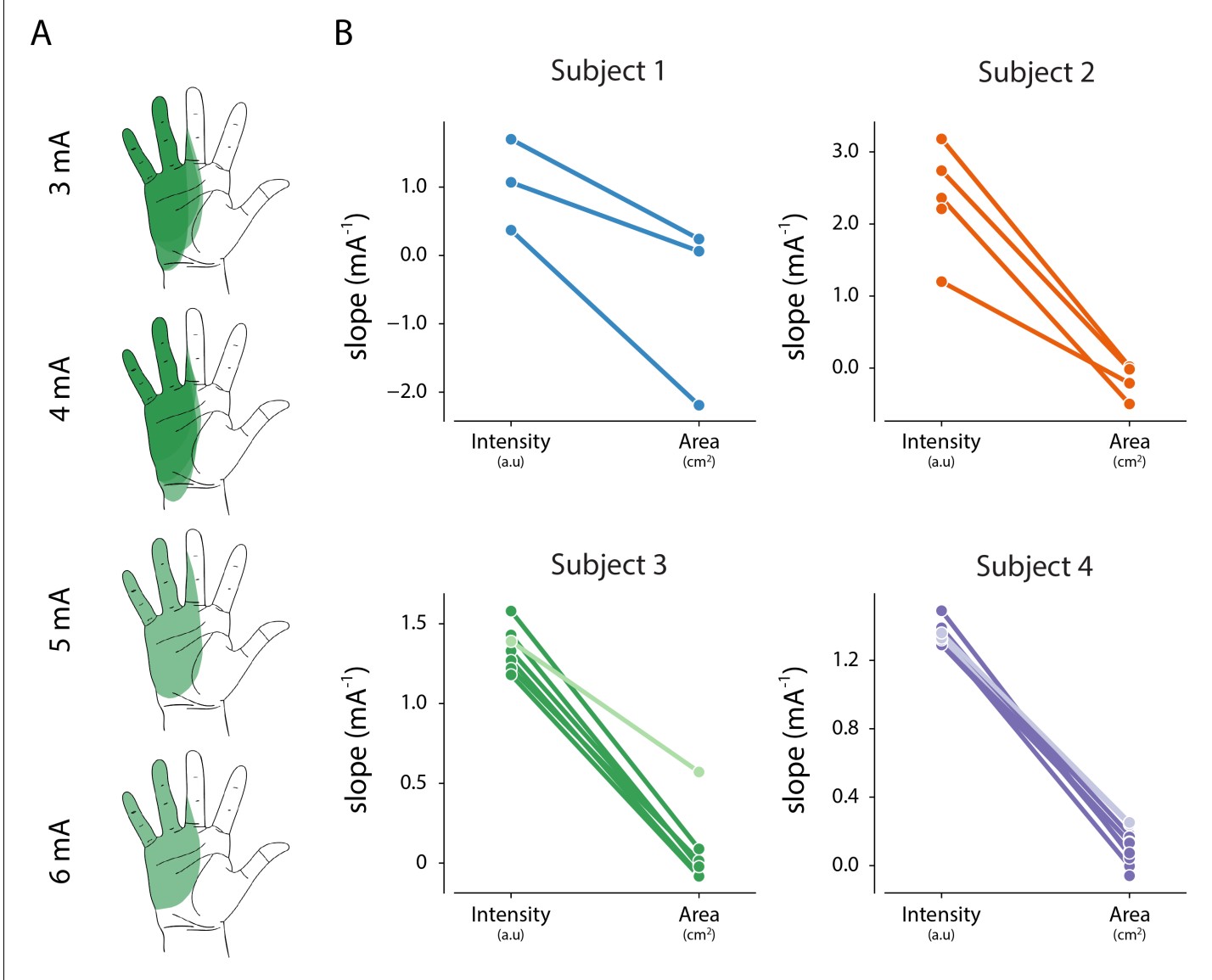

**Figure 5.** Relationship between intensity and area of evoked percept, and stimulation amplitude. (**A**) Example of the stability of percept area with increasing amplitude for one electrode in Subject 3. (**B**) Summary of the relationship between stimulation amplitude and percept characteristics for each electrode that evoked a percept in the phantom hand. The slope of the linear fit between stimulation amplitude and reported intensity was obtained from magnitude estimation trials. The slope of the linear fit between stimulation amplitude and percept area was obtained from percept mapping trials. Lighter shades represent electrodes where $p_{area} < 0.01$. The null hypothesis is that the slope of the linear fit is zero. Source data for panel B are available in *Figure 5—source data 1*.

The online version of this article includes the following source data for figure 5:

**Source data 1.** Intensity and area slope data for all percepts included in *Figure 5*.

fluoroscopy image and the final X-ray (*Figure 6A*) revealed that lead migration was largely restricted to the rostro-caudal axis. In all subjects, the largest caudal migration was observed when comparing the intraoperative fluoroscopy image with the X-ray at the end of first week (*Figure 6B*). One of the leads in Subject 2 almost completely migrated out of the epidural space in this post-operative period (*Figure 6B*), rendering it unusable for stimulation experiments. In contrast to the migration that occurred during the first week, X-rays from the first and last week of testing showed minimal lead migration (*Figure 6B*). In the weeks following the initial migration, the median migration in the rostro-caudal direction across the three leads in any subject never exceeded 5 mm. For all subjects,

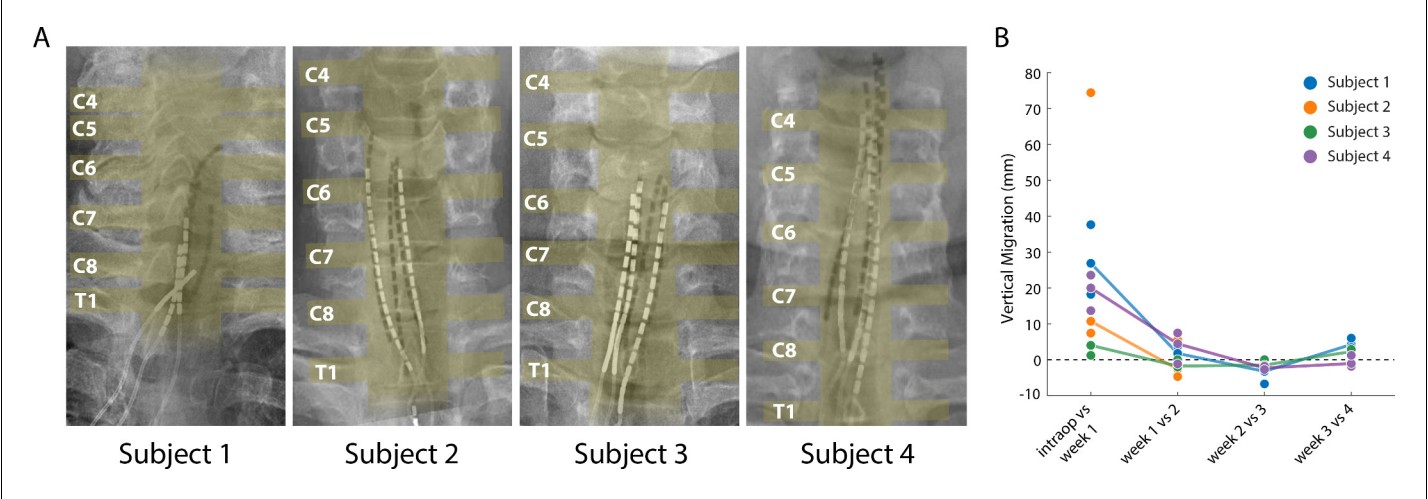

**Figure 6.** Stability of the SCS leads after implantation. (**A**) Composite image showing the changes in the position of the SCS leads in the epidural space. The intraoperative fluoroscopy image (contacts appear black) showing the position of the leads immediately after implantation is superimposed over the X-rays (contacts appear white) from week four for each subject. The labels on the left mark the dorsal root exiting at that level. The approximate location of the spinal cord and the roots is also shown in yellow overlay. For scale, each contact is 3 mm long. (**B**) Weekly migration of the rostral tip of each of the leads for the four subjects (blue, orange, green, and purple circles for Subjects 1–4, respectively). For week 1, the comparison was between the weekly X-ray and the intraoperative fluoroscopic image. For subsequent weeks, the comparison was done between the weekly X-ray and the one from the preceding week. Median migrations are shown (solid lines). The X-ray for Subject 2 was taken from week 2, before leads were explanted.

the initial placement of the leads rostral to the target cervical levels prevented loss of coverage of those spinal levels following the caudal migration of the leads.

We assessed the stability of each evoked percept throughout the duration of the study (e.g. *Figure 7A*) in terms of the threshold charge (*Figure 7B*) for evoking a percept in the missing hand. A one-way ANOVA performed for each subject confirmed that there was no significant difference in the threshold charge for each week for Subjects 1, 2, and 3 (p>0.01, F = 2.3, 1.1, 1.7 respectively). For Subject 4, there was a significant change in threshold after weeks one and three (p<0.01, F = 9.0). A post-hoc multiple pairwise comparison analysis using the Tukey HSD test confirmed that there was a significant increase in the thresholds between weeks one and two (p<0.01) and a significant decrease between weeks three and four (p<0.01).

We also characterized stability in terms of the size (area) and location (centroid) of percepts evoked in the missing hand. The centroid and area were calculated for all percepts evoked at the minimum stimulus amplitude that was tested at least once each week during the implant. If no stimulus amplitude was tested during every week of testing, the lowest stimulus amplitude that was tested for the next highest number of weeks (for at least two weeks) was chosen. We quantified the migration of the mean centroid location across all stimulus repetitions for each week with respect to the mean centroid location of the previous week for each electrode (*Figure 7C*). Across all subjects, the evoked percepts exhibited a median migration of 25.2 mm between weeks 1 and 2, 11.6 mm between weeks 2 and 3, and 9.9 mm between weeks 3 and 4. This week-to-week decrease in centroid migration follows the trend of decreased week-to-week vertical electrode migration (*Figure 7B*). Similarly, the change in area for each week was calculated with respect to the mean area of the previous week for each electrode (*Figure 7D*). Across all subjects, the median change in area of percepts evoked in the missing hand was 8.1 cm$^2$ between weeks 1 and 2, 0.14 cm$^2$ between weeks 2 and 3, and 1.1 cm$^2$ between weeks 3 and 4. We constructed two separate auto-regressive time series model to examine the changes in distributions of area and centroid distance over time, adjusting for autocorrelations in the data. We found a significant decrease in area over time across all weeks, (β = −0.201, p<0.01). For centroid migration, there was a decrease during weeks 2 (β = −23.224, p<0.05) and 3 (β = −40.585, p<0.01).

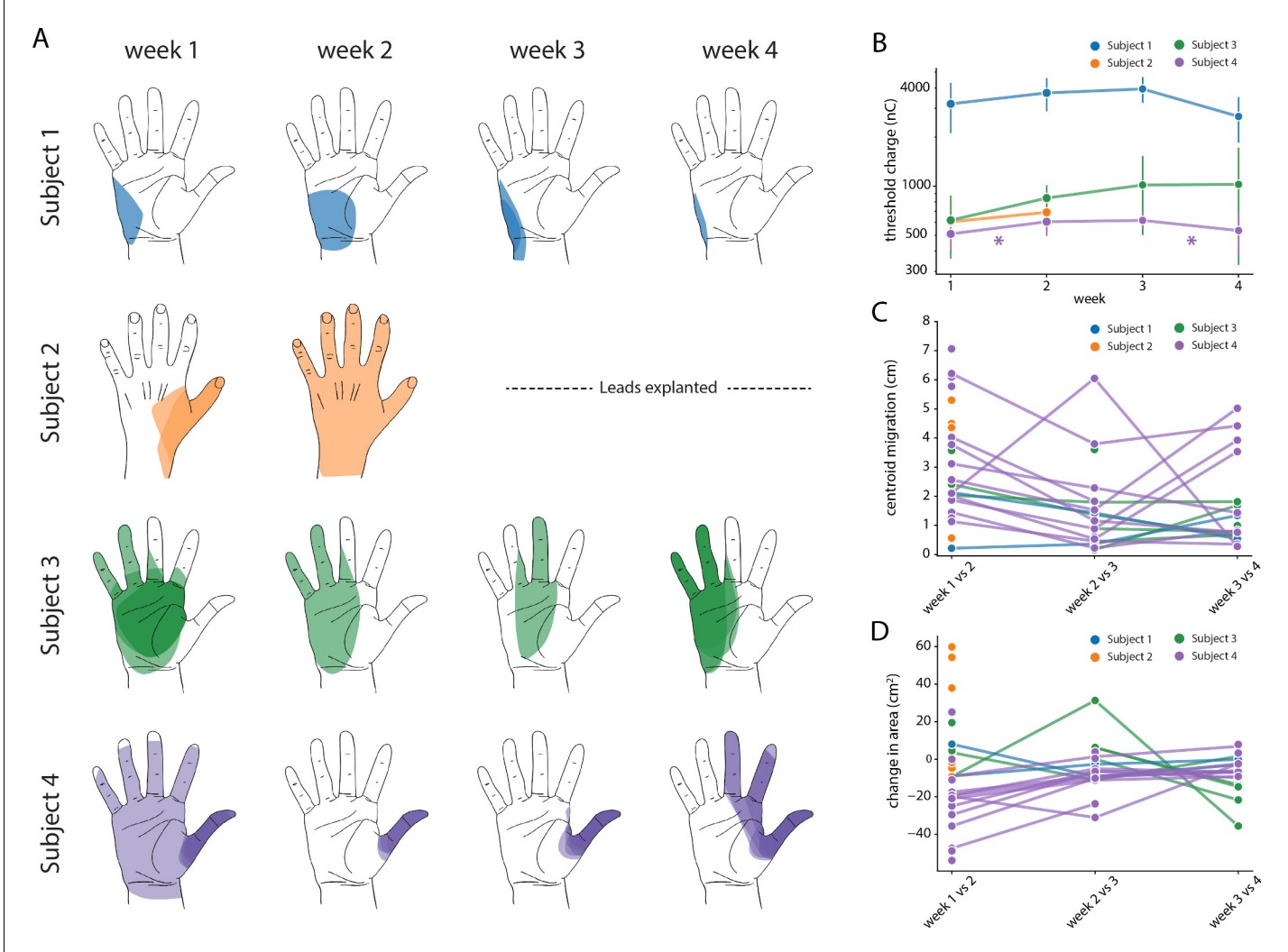

**Figure 7.** Stability of the sensory percepts. (**A**) Example sensory percepts from the hand for a single electrode in Subjects 1–4. For Subject 2, the percepts are shown for weeks 1 and 2 only, as the leads were explanted after that. The percepts shown were evoked by the minimum stimulus amplitude that was tested at least once per week for the maximal number of weeks. Each column shows the location of all percepts evoked for that week of testing. Multiple examples of the percepts evoked during the week are superimposed on each other as indicated by different shades. (**B**) Time course of the average stimulation threshold (in nC) for evoking a percept in the phantom hand for each subject. Weeks with a significant change in threshold are annotated with an asterisk. (**C**) Stability of the location of the evoked percept in the phantom hand. For each electrode, the centroid location of the evoked percept was compared between successive weeks. (**D**) Stability of the area of the evoked percept in the phantom hand. For each electrode, the area of the evoked percept was compared between successive weeks. Source data for panels B-D are available in *Figure 7—source data 1*, *Figure 7—source data 2*, and *Figure 7—source data 3*, respectively.

The online version of this article includes the following source data for figure 7:

**Source data 1.** Stimulation threshold charge for all electrodes included in *Figure 7B*.
**Source data 2.** Percept centroid migration distance data for all electrodes included in *Figure 7C*.
**Source data 3.** Change in percept area for all electrodes included in *Figure 7D*.

## Discussion

In this work, we show that epidural SCS has the potential to be an effective and stable approach for restoring sensation in people with upper-limb amputations. Further, we believe that percutaneous SCS can be used as an effective platform for development of somatosensory neuroprostheses, especially for labs that focus on advanced prosthetic control but have not developed their own

stimulation technologies. The electrodes used in this study are commercially available and were implanted using standard surgical techniques under local anesthesia during a procedure spanning 3–4 hr. In all subjects, we were able to evoke sensory percepts that were focal and localized to the distal missing limb. Critically, this included people with amputations ranging from trans-radial through shoulder disarticulation. The repertoire of sensory percepts elicited varied across subjects and thus, this approach would require user-dependent characterization, which is similar to the results reported by other studies of peripheral somatosensory neuroprostheses (*Charkhkar et al., 2018*; *Graczyk et al., 2018*; *Petrini et al., 2019c*; *Tan et al., 2015*). While most of the stimulation parameters evoked paresthesias, some of the percepts were more naturalistic. The intensity of the evoked sensations could be modulated by varying stimulation amplitude with little or no increase in the perceived area of the evoked sensations. Below we summarize these results and discuss their implications for the design of a somatosensory neuroprosthesis.

## Epidural SCS evokes sensations localized to the missing hand and arm

SCS-evoked sensory percepts were perceived to emanate from the missing limb in all subjects. Some percepts were highly localized to a single finger or focal region of the palm, while others were diffuse, covering large regions of the limb. In our second and third subjects, distal sensations were often accompanied by a secondary sensation at the residual limb. It is unclear whether these secondary sensations are a result of neuroplastic changes in the representation of the amputated hand in the cortex or are a limitation of the selectivity of the SCS electrodes used in this study. Future studies in people with intact limbs undergoing lateral SCS may help to differentiate these effects, since those subjects would not have similar neuroplastic changes. Factors that may limit stimulation selectivity with our approach include the thickness of the cerebrospinal fluid in the subdural space and the relatively large size of the contacts on the SCS leads. Consequently, the sensory percepts evoked in this study were sometimes more diffuse than those reported in other studies using peripheral neurostimulation approaches (*Charkhkar et al., 2018*; *Davis et al., 2016*; *Raspopovic et al., 2014*; *Tan et al., 2015*). Importantly though, they are comparable in focality to those used to effectively deliver sensory feedback during recent long-term take-home studies of bidirectional prosthesis using peripheral nerve stimulation (*Cuberovic et al., 2019*) or targeted reinnervation (*Schofield et al., 2020*).

In all except Subject 4, monopolar stimulation primarily evoked sensations in the forearm and upper arm, whereas multipolar stimulation allowed us to evoke sensations that were localized to distal regions of the missing hand and wrist. We could not identify any difference (e.g. in surgical technique) that led to this difference in focality of monopolar stimulation for Subject 4 as compared to all other subjects. In all subjects, the leads were steered toward the lateral spinal cord and spinal roots, ipsilateral to the amputation. At this location, the dorsal rootlets fan out under the dura before entering the spinal cord at the dorsal root entry zone. In the cervical spinal cord, the rootlets are each approximately 0.4–1.3 mm in diameter and densely packed with few spaces between them (*Alleyne et al., 1998*; *Karatas et al., 2005*; *Tanaka et al., 2000*). This arrangement, superficially resembling the flattened peripheral nerve cross-section achieved by the flat interface nerve electrode (*Charkhkar et al., 2018*; *Tan et al., 2015*), may lend itself to a higher degree of selective activation than could be achieved with stimulation of more traditional SCS targets such as the dorsal columns or the dorsal root ganglia, although this may require development of new SCS devices with more optimal electrode sizing and spacing.

The relationship between the locations of the electrodes and that of the evoked percepts showed marked inter-subject variability and deviation from expected dermatomes. For example, all electrodes in Subject 1 were in the T1 region, but the reported sensations were in the missing hand, a region covered by the C6–C8 dermatomes. Recently, it has been recognized that dermatomes inadequately reflect inter-individual variability in dermatome coverage and overlap, suggesting that the variability observed in our study may reflect natural inter-subject differences (*Lee et al., 2008*). However, a limitation of this study is that we did not directly image the spinal cord or dorsal roots. As such, we could not determine the exact spatial arrangement of the implanted SCS electrodes relative to target neural structures. Several research groups have developed highly detailed computational modeling techniques to study how the electric fields generated in SCS interact with neural structures (*Capogrosso et al., 2013*; *Greiner et al., 2020*; *Lempka et al., 2015*). These techniques could potentially help illuminate the specific neural targets and pathways that were activated in this

study. These observations combined with simulation studies could also inform the design of stimulation schemes and novel electrodes to improve the selectivity of our somatosensory neuroprosthesis.

## Stimulation parameters primarily modulate intensity of sensation

With respect to the perceptual qualities of evoked sensations in this study, we observed a robust relationship between stimulus amplitude and percept intensity. Every electrode tested across all four subjects demonstrated a statistically significant linear relationship between stimulation amplitude and perceived intensity. This is similar to what has been observed with peripheral nerve stimulation (*Graczyk et al., 2016*; *Petrini et al., 2019b*). Interestingly, we observed JNDs to be proportional to the stimulation amplitude with higher stimulation amplitudes resulting in larger JNDs. Such a relationship between JNDs and stimulus amplitude is consistent with Weber's law which governs the behavior of most peripheral sensory receptors (*Gös, 1959*). We also observed that subject responses could be separated into three, but not five, intensity categories (i.e. low, medium, and high) based on the stimulation amplitude, which suggests that they would be able to successfully perform a 3-level discrimination task based only on perceived intensity, such as identifying three different force levels exerted by objects of different stiffnesses. These expected performance levels are similar to the success rates for object stiffness experiments demonstrated by others using peripheral nerve stimulation to restore somatosensation (*Raspopovic et al., 2014*), and it is possible that they could improve with time, training, continuous (rather than discrete) modulation of amplitude, and the addition of active efferent control of a prosthesis. Future work should focus on demonstration of such closed-loop control with sensory feedback using lateral SCS.

An increase in stimulus amplitude is thought to increase perceived intensity by recruiting a larger volume of somatosensory afferent neurons (*Graczyk et al., 2016*). An increase in the volume of recruited neurons could also result in an increase in percept area. However, we observed little to no effect of increasing stimulation amplitude on percept area. It is possible that the anatomical distance between adjacent spinal roots reduces this effect (*Greiner et al., 2020*). Additionally, it is currently unknown whether there is strong somatotopic organization of the fanned-out dorsal rootlets where they enter the spinal cord (i.e. whether neighboring rootlets innervate neighboring patches of skin). The minor changes in focality of sensation as amplitude increases may be due to the presence of this somatotopic organization and recruitment of neurons in adjacent rootlets.

A primary aim of providing artificial somatosensory feedback has been to evoke naturalistic sensations, particularly those described as touch or pressure. Most of the percepts reported in this and previous studies of somatosensory neuroprostheses have been described as paresthesias. However, for Subjects 2 and 3, we report that 8.25% and 19.5% of all evoked percepts were described as touch or pressure alone. Studies that relied on peripheral nerve stimulation to restore somatosensory feedback have reported similar proportions of naturalistic percepts, e.g. 8–30% as touch-like percepts (*Petrini et al., 2019b*; *Strauss et al., 2019*; *Tan et al., 2014*) and 2–29% as pressure-like percepts (*D'Anna et al., 2019*; *George et al., 2019*; *Petrini et al., 2019b*; *Strauss et al., 2019*). Continuous modulation of stimulus parameters, such as modulating pulse width (*Charkhkar et al., 2018*; *Tan et al., 2014*) or varying charge density [*Charkhkar et al., 2018*] have been proposed to evoke more naturalistic cutaneous or proprioceptive sensations. However, a recent study demonstrated that patterned stimulation did not reliably change paresthetic sensations to more naturalistic ones (*Ortiz-Catalan et al., 2019*). Additionally, biomimetic stimulus trains [*George et al., 2019*; *Okorokova et al., 2018*; *Valle et al., 2018*] have been proposed to evoke more naturalistic sensations, though none of these approaches have established a stimulation paradigm that reliably elicits naturalistic sensations across subjects. We too, did not uncover a reliable way to evoke naturalistic sensation during the course of this study. Thus, the choice of electrodes and stimulation parameters would have to be optimized for each individual user to evoke percepts with the desired modalities (*Ortiz-Catalan et al., 2019*).

Only one subject (Subject 4, trans-radial amputation) reported proprioceptive percepts that were repeatedly evoked over more than a few minutes. This result aligns with other studies, which often report only limited examples of proprioceptive percepts (*George et al., 2019*), or which describe proprioceptive sensations that result directly from muscle contractions in the residual limb (*Petrini et al., 2019a*). While SCS did not evoke overt reflexive movements of the residual limb in any subject at the stimulation amplitudes used in this study, it is possible that these proprioceptive percepts result from small reflexive contractions of residual limb muscles which themselves activate

muscle spindle afferents. Complex coordination of activation of muscle spindle and cutaneous (e.g. slowly adapting type II) afferents may be required for directly evoking realistic kinesthetic percepts. Future work should explore the downstream effects of stimulation of proprioceptive and cutaneous afferents on perception of kinesthesia. Regardless, we propose that even though we evoked primarily paresthetic sensations, the ability to evoke these percepts via a clinically translatable approach in individuals with high-level amputations establishes the promise of this approach towards restoring sensation.

## Percutaneous SCS electrodes and evoked percepts are stable over one month

The location of the implanted SCS electrodes and the corresponding evoked percepts showed only minor migration across the duration of implantation. In clinical practice, SCS lead migration is a common complication, occurring in as many as 15–20% of cases (*Cameron, 2004*; *Kinfe et al., 2014*; *Mekhail et al., 2011*; *Mironer et al., 2004*), and is typically classified by a complete loss of paresthetic coverage of the region of interest. Repeated monitoring of both the physical location of the SCS leads and the evoked sensations demonstrated that there was some migration immediately after implantation, but minimal movement thereafter. As a preemptive measure against loss of coverage due to the initial migration, we opted to use 16-contact leads in Subjects 2–4. By placing the leads such that the most rostral contacts were above the target spinal levels, we ensured continued coverage even in the case of caudal migration. It is worth noting that we did not anchor these leads to any bony structures or nearby tissue. Future permanently implanted systems for restoring sensation using SCS can utilize these anchoring techniques and thereby reduce or eliminate lead migration (*Mekhail et al., 2011*). The stability in the electrodes is reflected in the stability of the evoked percepts. In the hand region, we observed a migration of evoked percepts of 10–25 mm, which is similar to the shift reported in peripheral stimulation approaches (*Tan et al., 2015*). Moreover, given that the spatial acuity in the palm region is approximately 8–10 mm (*Catley et al., 2013*; *Craig and Lyle, 2001*; *Solomonow et al., 1977*; *Tong et al., 2013*), the scale of migration observed is within the range that would not likely be detectable by the user.

## Comparison to alternative approaches

The techniques described in this study have both important advantages and disadvantages that should be considered when selecting an approach for restoring sensation after upper-limb amputation. A major advantage of the percutaneous approach described here is that there is a relatively low barrier to initiating clinical studies because the electrodes are commercially available from multiple manufacturers and the surgical procedures are commonly performed at most major medical centers. However, this reliance on commercially available electrodes also likely limited selectivity and focality. Regardless, a great deal of technical and scientific development can be achieved with this approach before moving on to more complex studies involving custom electrodes and implantable stimulators.

Another major advantage of the approach is its viability for people with high-level amputations, in which the peripheral nerve has been amputated. In this population, the spinal cord and roots typically remain intact, and we have demonstrated that stimulation of those structures can produce focal sensations in the missing hand. Currently, the only other viable neuroprosthetic techniques for restoring sensation after proximal amputation are invasive approaches such as targeted reinnervation or stimulation of structures in the central nervous system.

As compared to other techniques that focus on peripheral nerve stimulation, such as epineural stimulation with cuff electrodes or penetrating stimulation with Utah arrays or longitudinal intrafascicular electrodes, our results demonstrate substantially less focal percepts and less consistent coverage of each individual digit across subjects. Further, sensations in the hand were often accompanied by a sensation on the residual limb. It is currently unclear to what degree this is a limitation of the relatively large size of the electrodes we used here, as opposed to a fundamental limitation of the selectivity of epidural SCS. Future work will focus on computational studies to explore this question and design new electrodes that can more selectively target the sensory afferents in the dorsal rootlets to maximize the selectivity and focality of our approach. With respect to clinical applications, the 3-fold dynamic range afforded by the stimulus amplitude is similar to those reported previously

(*Petrini et al., 2019b*). Though the absolute current values we used are an order of magnitude higher than those required for peripheral nerve stimulation, epidural stimulation systems are widely used in a clinical setting and also in patient homes. This suggests that a neuroprosthetic device based on this approach can be effectively utilized in clinical or home setting.

An important limitation of this study is that we focused on characterizing the sensations evoked by SCS but did not demonstrate that those sensations could be used as part of a closed-loop neuro-prosthetic system. While we demonstrate that many of the qualities of the evoked sensations are similar to those reported by others (e.g. sensation intensity modulates linearly with stimulation amplitude), it will be critical to demonstrate that sensations remain stable and are useful during closed-loop prosthetic applications. For example, while we did not control subject posture during any of our experiments, it will be important to demonstrate that sensations remain stable during intentional movements of the neck, shoulders, and arms. Certainly, future work will focus on achieving these demonstrations and characterizing the effects of sensory restoration via SCS on dexterous control of prosthetic limbs.

## Conclusions

Since this approach targets proximal neural pathways, SCS-mediated sensory restoration lends itself to use for a wide range of populations, such as individuals with proximal amputations and those with peripheral neuropathies in which stimulation of peripheral nerves may be difficult or impossible. Provided that the injury does not affect the dorsal roots and spinal cord, our results suggest that these techniques can be effective in restoring sensation, regardless of the level of limb loss. Moreover, the widespread clinical use of SCS and the well-understood risk profile provide a potential pathway towards clinical adoption of these techniques for a somatosensory neuroprosthesis.

# Materials and methods

## Study design

The aim of this study was to investigate whether electrical stimulation of lateral structures in the cervical spinal cord could evoke sensations that are consistently perceived to emanate from the missing hand and arm. We also aimed to characterize those sensations and establish the relationship between stimulation parameters and the perceptual quality of evoked sensory percepts. Four subjects with upper-limb amputations (three females, one male; *Table 1*) were recruited for this study. Three amputations were between the elbow and shoulder and one was below the elbow. The time since amputation ranged from 2 to 16 years. All procedures and experiments were approved by the University of Pittsburgh and Army Research Labs Institutional Review Boards and subjects provided informed consent before participation.

## Electrode implantation

SCS leads were implanted through a minimally invasive, outpatient procedure performed under local anesthesia. With the subject in a prone position, three 8- or 16-contact SCS leads (Infinion, Boston Scientific) were percutaneously inserted into the epidural space on the dorsal side of the C5–C8 spinal cord through a 14-gauge Tuohy needle. Contacts were 3 mm long, with 1 mm inter-contact spacing. Leads were steered via a stylet under fluoroscopic guidance, and electrode placement was iteratively adjusted based on the subjects' report of the location of sensations evoked by intraoperative stimulation. The entire procedure usually took approximately 3–4 hr. The leads were maintained for up to 29 days and subsequently explanted by gently pulling on the external portion of the lead. Subjects attended testing sessions 3–4 days per week during the implantation period. The testing sessions lasted up to a maximum of 8 hr. Lead location and migration were monitored via weekly coronal and sagittal X-rays throughout the duration of implant.

## Neural stimulation

During testing sessions, stimulation was delivered using three 32-channel stimulators (Nano 2+Stim; Ripple, Inc). The maximum current output for these stimulators was 1.5 mA per channel. In order to achieve the higher current amplitudes required for SCS, a custom-built circuit board was used to short together the output of groups of four channels, thereby increasing the maximum possible

output to 6 mA per channel resulting in a total of 8 effective channels per stimulator. Custom adapters were used to connect each stimulator to eight contacts on each of the implanted leads. Custom software in MATLAB was used to trigger and control stimulation.

Stimulation pulse trains were charge-balanced square pulses, with either asymmetric or symmetric cathodic and anodic phases. For Subjects 1-3, the first phase of stimulation was cathodic, while for Subject 4, an error in the stimulation control code caused the first phase of stimulation to be anodic. For asymmetric pulses, the second phase was twice the duration and half the amplitude of the first phase. Stimulation was performed either in a monopolar configuration, with the ground electrode placed at a distant location such as on the skin at the shoulder or hip, or in a multipolar configuration with one or more local SCS contacts acting as the return path. Stimulation frequencies and pulse widths ranged from 1 to 300 Hz and 50–1000 µs, respectively. The interphase interval was 60 µs. All stimulus amplitudes reported in this manuscript refer to the first phase amplitude.

## Recording perceptual responses

The first few sessions of testing were primarily devoted to recording the location and perceptual quality of sensory percepts evoked with various stimulation configurations. An auditory cue was provided to denote the onset of stimulation. At the offset of each stimulation train, the subject used a touchscreen interface developed in Python (*Figure 1—figure supplement 2*) to document the location and perceptual quality of the evoked sensation. This interface can be downloaded from *Nanivadekar et al., 2020* https://github.com/pitt-rnel/perceptmapper. The location of the sensory percept was recorded by the subject using a free-hand drawing indicating the outline of the evoked percept on an image of the appropriate body segment (i.e., hand, arm, or torso). The percept quality was recorded using several descriptors: mechanical (touch, pressure, or sharp), tingle (electrical, tickle, itch, or pins and needles), movement (vibration, movement across skin, or movement of body/limb/joint), temperature, pain due to stimulation, and phantom limb pain. Each descriptor had an associated scale ranging from 0 to 10 to record the corresponding perceived intensity. Additionally, the subject was instructed to rate the naturalness (0–10) and the depth of the perceived location of the percept (on or below the skin, or both). This set of descriptors have been used previously to characterize evoked sensory percepts (*Heming et al., 2010*; *Lenz et al., 1993*). The order of stimulation electrodes and amplitudes was randomized to prevent subjects from predicting the location and perceptual qualities of sensations from previous trials. All percepts that were localized ipsilateral to the amputation were included for analysis in this work. In *Figure 1*, only those percepts which show less than 70% area overlap (as in *Charkhkar et al., 2018*) with any other percept are shown for clarity. *Supplementary file 1* visualizes all the evoked percepts in an interactive fashion.

## Analyzing sensory percept distribution

For each trial, subjects were allowed to report more than one descriptor simultaneously. Each unique combination of 'mechanical', 'movement', and 'tingle' descriptors was considered a separate modality for the evoked percept. All percepts that contained a descriptor for tingle ('electric current', 'tickle', 'sharp', 'pins and needles') were considered paresthetic and were grouped together. A sunburst plot was constructed for each subject to analyze the fraction of paresthetic and non-paresthetic percepts that contained mechanical or movement components. Therefore, all unique modalities were divided in to three groups: paresthetic percepts that had 'tingle' but no 'mechanical' or 'movement' component (*Figure 3*, teal sectors), mixed percepts that had a 'mechanical' and/or 'movement' component and 'tingle' (*Figure 3*, grey sectors), and non-paresthetic percepts that only had 'mechanical' and/or 'movement' components (*Figure 3* red sectors). For each sunburst plot, the inner, middle, and outer annuli represent 'tingle', 'mechanical', and 'movement' modality descriptors, respectively. Each sector represents a unique descriptor and the size of each sector represents the fraction of all percepts that contained the corresponding descriptor. This allows us to identify the distribution of unique modalities, such that, for a given sector in the tingle annulus (for example, n = 761 for Subject 2, 'tingle'), we can identify the fraction of percepts that had a specific mechanical descriptor (e.g. 'sharp' n = 245) and the fraction of these percepts that had a specific movement descriptor (e.g. 'vibration' n = 104). An interactive version of *Figure 3* with expandable sectors for each descriptor is available in *Supplementary file 3*.

The spinal cord segment targeted by stimulation through each electrode was inferred from the X-ray images. We used the pedicles of each vertebra to mark the boundaries that separated each spinal root (*Figure 2B*). These boundaries provided an anatomical marker to establish where each electrode was located, in the rostrocaudal axis. Similarly, boundaries were drawn on the body segment outline images to divide them into seven anatomical segments (*Figure 2A*) including thumb, D2–D3, D4–D5, wrist, forearm, elbow, and upper arm. The sensory percepts were categorized as being associated with one of the seven anatomical segments based on which segment contained the maximal area of the perceived sensation. For this analysis, only those electrodes that evoked a sensory percept ipsilateral (n = 315) to the amputation were included. Electrodes that only evoked bilateral (n = 64) and contralateral (n = 68) sensations at threshold would not be useful for neuroprosthetic applications for people with unilateral amputation and were excluded. Dermatome maps were generated per subject, by determining the proportion of electrodes situated at each spinal level that evoked a sensation in a specific anatomical region.

## Quantifying lead and percept migration

The intraoperative fluoroscopy image, superimposed over the X-rays from the first and last week of testing, gave an indication of gross movements of the leads. Using bony landmarks, the X-ray from the first week was aligned to the intraoperative fluoroscopy image, and each subsequent X-ray was aligned to the X-ray from the previous week using an affine transformation method in MATLAB. The SCS contact that appeared to be most parallel to the plane of imaging was used to determine the scale length for the image (SCS contacts are 3 mm in length). For each lead, the distance between the rostral tips of the electrodes as seen in the aligned image pairs (*Figure 6*) was measured to determine the rostro-caudal migration. Positive values signified caudal migration and negative values signified rostral migration.

For all electrodes that evoked a percept in the missing hand, the threshold charge was calculated for each week. A one-way ANOVA was performed for each subject to test for differences in thresholds across weeks. For subjects where a significant difference was reported, a post-hoc multiple pairwise comparison analysis using the Tukey HSD was performed to identify the pairs of consecutive weeks with a significant difference in thresholds. To quantify migration of perceived sensations, we measured the change in the position of the centroid and the change in area of each percept that was localized to the hand. For sensations that included a percept outside the hand, we only used the hand percept in these calculations, as this is the most relevant location for a somatosensory neuroprosthesis. We chose the minimum stimulus amplitude that was tested at least once per week for the highest number of weeks during the implant. We quantified the migration of the mean percept centroid for each week, with respect to the mean percept centroid for the previous week. This analysis was repeated for all electrodes. Similarly, to quantify the change in percept area, the mean area of the percept for each week was compared to the mean area for the previous week. The distances were converted to millimeters using the average hand length of 189 mm (as measured from the tip of the middle finger to the wrist) and average palmar area of 75 cm$^2$ of a human male (*Agarwal and Sahu, 2010*; *Ilayperuma et al., 2009*; *Kono et al., 2014*; *Martin and Nguyen, 2004*; *Rhodes et al., 2013*; *Zafar et al., 2017*). All electrodes that were tested in at least two of the weeks were included in the analysis.

We also constructed separate auto-regressive time series models to examine the changes in distributions for both area and centroid migration over time, adjusting for autocorrelations in the data. The AUTOREG procedure in SAS estimates and forecasts linear regression models for time series data when the errors are autocorrelated or heteroscedastic. If the error term is autocorrelated (which occurs with time series data), the efficiency of ordinary least-squares (OLS) parameter estimates is adversely affected and standard error estimates are biased, thus the autoregressive error model corrects for serial correlation. For models with time-dependent regressors, the, AUTOREG procedure performs the Durbin t-test and the Durbin h-test for first-order autocorrelation and reports marginal significance levels.

## Detection thresholds

A two-alternative forced choice task was used to determine detection thresholds. The subject was instructed to focus on a fixation cross on a screen. Two one-second-long windows, separated by a

variable delay period, were presented and indicated by a change in the color of the fixation cross. Stimulation was randomly assigned to one of the two windows. After the second of the two windows, the fixation cross disappeared, and the participant was asked to report which window contained the stimulus. The stimulus amplitude for each trial was varied using a threshold tracking method (*Leek, 2001*; *Levitt, 1971*) with a 'one-up, three-down' design. In this design, an incorrect answer resulted in an increase in stimulus amplitude for the next trial while three consecutive correct trials were required before the stimulus amplitude was decreased. Stimulus amplitude was always changed by a factor of 2 dB. Five changes in direction of the stimulus amplitude, either increasing to decreasing or vice versa, signaled the end of the task. Using this task design, the detection threshold was determined online as the average of the last 10 trials before the fifth change in direction. A detection threshold calculated this way corresponds approximately to correctly identifying the window containing the stimulus 75% of the time (*García-Pérez, 1998*). To get a finer estimate of the detection threshold we also used a non-adaptive design in a subset of trials for Subject 4, where we presented a predetermined set of stimulus amplitudes. This block of stimulus amplitudes was repeated up to eight times and the presentation sequence was randomized within each block. A cumulative-normal psychometric curve was fit to both types of detection experiments *post-hoc* using the Palamedes toolbox (*Kingdom and Prins, 2016*) with the guessing rate $\gamma$ and lapse rate $\lambda$ held fixed at 0.5 and 0 respectively. The detection threshold was calculated as the stimulus amplitude at the 75% accuracy level. Tasks in which accuracy levels for all stimulus amplitudes were <0.6 or >0.9 were omitted from this analysis. We carried out a goodness-of-fit analysis with 1000 simulations using the Palamedes toolbox and discarded any fit with probability of transformed likelihood ratio (pTLR) less than 0.05. pTLR signifies the proportion of simulated likelihood ratios that were smaller than the likelihood ratio obtained from the data and it spans 0–1 with a higher value signifying a better fit and values below 0.05 signifying an unacceptable fit. Thresholds calculated for the same electrodes on different days were averaged together to obtain a mean detection threshold for each electrode, with all other stimulus parameters (e.g. frequency, pulse width) held constant.

## Just-noticeable differences

A similar two-alternative forced choice task was used to determine just-noticeable differences (JND) for stimulation amplitude. The design of the task was identical to the detection task except stimulation was provided in both the windows and the subject was instructed to choose the window where the stimulus was perceived as being at a higher intensity. One of the stimulation amplitudes in every trial was held constant while the other was chosen randomly from a list of stimulus amplitudes constituting a block. The constant amplitude was either fixed at 2.5 mA for the lower standard amplitude or at 4.0 mA for the higher standard amplitude. The windows in which standard and the test amplitude were administered was randomized as well. This block of stimulus amplitudes was repeated up to eight times and the presentation sequence was randomized within each block. A cumulative-normal psychometric curve was fit to the data post-hoc using the Palamedes toolbox (*Kingdom and Prins, 2016*) with the guessing rate $\gamma$ and lapse rate $\lambda$ held fixed at 0.5 and 0 respectively. The JND was calculated as the stimulus amplitude at the 75% accuracy level. Tasks in which accuracy levels for all stimulus amplitudes were <0.6 or >0.9 were omitted from this analysis. We carried out a goodness-of-fit analysis with 1000 simulations using the Palamedes toolbox and discarded any fit with pTLR <0.05. To determine average JNDs at the two different standard amplitudes, we included data from only those electrodes for which testing at both standard amplitudes were carried out in the same session. JNDs calculated for the same standard amplitude on different electrodes were averaged together to obtain a mean JND for each standard amplitude. As JNDs were expected to be highly subject-specific, data from different subjects were not pooled together.

## Perceived intensities of the evoked sensory percepts

A free magnitude estimation task was used to determine the relationship between stimulus amplitude and perceived intensity of the evoked sensations (*Ellermeier et al., 1991*; *Stevens, 1986*; *Stevens, 1956*; *Verrillo et al., 1969*). In this task, subjects were instructed to rate the perceived intensity on an open-ended numerical scale as stimulation amplitude was varied randomly. A block of stimulus amplitudes consisted of 6–10 values linearly spaced between the detection threshold of the electrode being tested and the highest value that did not evoke a painful percept up to 6 mA.

This block of chosen amplitudes was presented six times and the presentation sequence was randomized within each block. The subject was instructed to scale the response appropriately such that a doubling in perceived intensity was reported as a doubling in the numerical response. Zero was used to denote that no sensation was perceived in response to the stimulus. During the first block, the subject experienced the full range of stimulation amplitudes while establishing their subjective scale, so data from this block were not included in the analysis. Data across electrodes or across different testing sessions were compared after normalizing each electrode to its mean response.

We performed a *post-hoc* analysis to determine the maximum number of intensities a subject would likely be able to discriminate. For each electrode, stimulation amplitude was normalized to the maximum amplitude tested and the range of stimulation amplitude was partitioned into three or five linearly spaced discrete values. Similarly, the perceived intensities reported by the subjects were normalized to the maximum reported intensity and partitioned into three or five discrete linearly spaced ranges. Across all subjects, the distribution of the binned reported intensity for each discretized stimulation level was used to estimate how reliably subjects would be able to distinguish feedback at three or five different amplitudes (*Figure 4F* and *Figure 4—figure supplement 2*).

To determine whether stimulation amplitude had differential effects on the area and intensity of evoked percepts, we computed the least-squares regression line for the relationship between stimulation amplitude and percept intensity and from magnitude estimation trials and stimulation amplitude and percept area from percept mapping trials. The two-side p-values ($p_{int}$ and $p_{area}$, respectively) for each line were obtained for the null hypothesis that the slope of the regression line was zero and the slopes of the two lines were compared for each electrode. Instances where the slopes of each line were significantly different indicate electrodes where stimulation amplitude can modulate percept intensity independent of the area of the percept.

## Acknowledgements

We would like to thank our study participants for their extraordinary commitment to this study, their patience with the experiments, and the deep insights provided by them; the clinicians and researchers at University of Pittsburgh; H Stein, L Wilcox, E Bird and B Bigelow for their recruitment efforts, regulatory compliance and clinical scheduling; H Jourdan for organizational support.

## Additional information

### Funding

| Funder | Grant reference number | Author |
|---|---|---|
| Defense Advanced Research Projects Agency | W911NF-15-2-0016 | Santosh Chandrasekaran<br>Ameya C Nanivadekar<br>Eric R Helm<br>Michael L Boninger<br>Jennifer Collinger<br>Robert Gaunt<br>Lee E Fisher |

The funder provided monetary resources for the project and supervision via required reports, milestones, and deliverables.

### Author contributions

Santosh Chandrasekaran, Conceptualization, Data curation, Software, Formal analysis, Validation, Investigation, Visualization, Methodology, Writing - original draft, Writing - review and editing; Ameya C Nanivadekar, Resources, Data curation, Software, Formal analysis, Validation, Investigation, Visualization, Writing - original draft, Writing - review and editing; Gina McKernan, Formal analysis, Validation, Writing - review and editing; Eric R Helm, Conceptualization, Funding acquisition, Methodology, Writing - review and editing; Michael L Boninger, Conceptualization, Resources, Supervision, Funding acquisition, Project administration, Writing - review and editing; Jennifer L Collinger, Conceptualization, Supervision, Funding acquisition, Methodology, Project administration, Writing - review and editing; Robert A Gaunt, Conceptualization, Resources, Supervision, Funding acquisition,

Methodology, Project administration, Writing - review and editing; Lee E Fisher, Conceptualization, Resources, Supervision, Funding acquisition, Investigation, Methodology, Writing - original draft, Project administration, Writing - review and editing

### Author ORCIDs
Santosh Chandrasekaran ⑩ https://orcid.org/0000-0002-9795-119X
Ameya C Nanivadekar ⑩ https://orcid.org/0000-0001-7889-9077
Gina McKernan ⑩ https://orcid.org/0000-0003-3234-7802
Eric R Helm ⑩ https://orcid.org/0000-0002-7316-4941
Michael L Boninger ⑩ https://orcid.org/0000-0001-6966-919X
Jennifer L Collinger ⑩ https://orcid.org/0000-0002-4517-5395
Robert A Gaunt ⑩ https://orcid.org/0000-0001-6202-5818
Lee E Fisher ⑩ https://orcid.org/0000-0002-9072-3119

### Ethics
Clinical trial registration NCT02684201.
Human subjects: All procedures and experiments were approved by the University of Pittsburgh and Army Research Labs Institutional Review Boards (Protocol #: STUDY19100220) and subjects provided informed consent before participation.

### Decision letter and Author response
Decision letter https://doi.org/10.7554/eLife.54349.sa1
Author response https://doi.org/10.7554/eLife.54349.sa2

## Additional files

### Supplementary files
• Supplementary file 1. An interactive version of *Figure 1* showing all percepts included in analyses in this study.

• Supplementary file 2. Summary of psychophysics testing for each subject. For detection and discrimination trials the threshold (TH) and JND per stimulation channel are listed along with the corresponding frequency and pulse width that were used.

• Supplementary file 3. An interactive version of *Figure 3*. Clicking on individual sectors provides additional detail.

• Transparent reporting form

### Data availability
All data generated or analyzed during this study are included in the manuscript and supporting files. We have included a supplementary interactable html file that allows readers to explore all of the sensations that were evoked in the missing limb in all subjects and another file that allows readers to explore the modality of sensations evoked in all subjects.

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
