## [Decision Letter]

**Acceptance summary:**

This paper provides a transparent and balanced communication of the opportunities and disadvantages afforded by spinal cord stimulation for sensory restoration. We are excited about the opportunities offered to the community by the sensations reporting tool, which is of particular value considering the overly descriptive and non-uniform reporting standards in the community. This tool will provide a much needed means for quantitative comparison of results across studies and interventions.

**Decision letter after peer review:**

Thank you for sending your article entitled "Sensory restoration by epidural stimulation of dorsal spinal cord in upper-limb amputees" for peer review at *eLife*. Your article has been evaluated by three peer reviewers, one of whom is a member of our Board of Reviewing Editors, and the evaluation is being overseen by Richard Ivry as the Senior Editor.

The reviewers and editors agreed that your manuscript provides an interesting approach for sensory restoration that needs to be documented, and appreciated the efforts to characterize the feasibility of this procedure. However, they also agreed that there are too many serious issues in the manuscript that presently preclude the translational impact of the study. To give three examples:

1) Very limited sample size, combined with inherent inter-subject variability deems the interpretation of the findings very challenging.

2) Locality: How focal is the stimulation, and is it useful for the purpose of touch localisation? And how can one deal with the double sensations (on the residual limb)?

3) Action: What are the consequences of the stimulation on motor control and vice versus

Considering the limited translational or neuroscience impact presently afforded by the manuscript, the main innovation the paper provides is methodological. And here we do see considerable value in the study. A key strength of the paper is that it reveals the challenges of the technique, as well as the opportunities.

As such, we would welcome a revision that is restructured as a submission under *eLife*'s Tools and Resources category. https://elifesciences.org/articles/07083. Note that this will require that you revise some parts of the manuscript to better reflect some methodological aspects (for example, what challenging decisions they needed to make during surgical insertion, what worked and didn't work for the stimulation, etc) and/or share some of their code (e.g. the tool they developed to capture the sensory percepts). A revision for the Tools and Resources category would also provide an opportunity to provide a more balanced account of the feasibility of this technique for sensory substitution. This perspective could be worked more explicitly into the manuscript (the reviewers made multiple suggestions to help guide the revisions).

*Reviewer #1:*

The authors demonstrate that epidural spinal cord stimulation in amputees evokes sensations that are perceived as emanating from the missing limb, and hand in particular, providing an interesting candidate procedure for restoring the sense of touch for prosthesis usage. While previous studies have reported that SCS evokes sensations, this is the first study to meticulously characterise the feasibility of this procedure for sensory restoration in upper limb amputees. The authors demonstrate that by manipulating the stimulation attributes they are able to linearly modulate the resulting sensations, with relatively high predictability of perceptual outcomes. While other techniques are currently available to surgically restore tactile sensations, the current approach provides unique advantages, both in terms of surgical practicality and with respect to amputees with proximal injuries. The authors did a commendable job in characterising stimulus-response profiles, which is no small task considering the massive parameter space and difficulty of measuring subjective reports. Although preliminary, I believe this effort will provide an important first point of reference for considering further the feasibility of this approach.

While there was a lot to like in this study, there were also some major limitations that need to be considered. First, the spatial stability of the stimuli over weeks was pretty disappointing. Is it true that this is comparable with the current state of the art for peripheral stimulation techniques? Second, as emphasised in the manuscript, the inter-individual differences are massive. Because the study only involves 4 subjects, it’s hard to determine what is driving the differences between the subjects, which might be very important for considering the translational power of this technique (e.g. specificity of stimulation site). Third, while this approach is designed for improving prosthesis usage, the current design/results do not seem to take into consideration the practical considerations of prosthesis usage (e.g. active movements, spatiotemporal profile of tactile feedback). For all these reasons, based on the evidence presented here, I'm not convinced that this approach is entirely feasible/promising for tactile restoration of touch. This is not to say that the manuscript should not be published – this is conceptually a very appealing idea and it’s important to set the record clear on the advantages and limitations that it entails. But the manuscript, and Abstract and Discussion in particular, should be changed to reflect these limitations better, to produce a more balanced perspective on the presented results.

1) Beyond the translational significance of this manuscript, it offer modest innovation for basic neuroscience (e.g. the observation on the somatotopy across dermatomes). So it might suit better the methodological format Tools and resources? Note that this will require the authors to share some of their code (e.g. for capturing the sensory percepts).

2) Practically, for the purpose of tactile feedback for prosthesis usage, is this procedure producing sufficient spatial resolution? In other words, are the percepts focal enough, and spatially synchronised sufficiently? Are these percept impacted when the subject is mobile?

3) The authors show very impressive performance with the bayesian classifier for predicting the categorical description, but could this be put into better spatio-temporal context, relating to the other key attributes of the percept? Also, could they correctly classify the sensations across sessions/weeks?

4) The statistical analysis requires further details. What were the parameters used for the GLM? How were they modelled (e.g. repeated model, fixed, etc)? authors should be weary of collapsing data across electrodes and participants (fixed effect, e.g. Figure 3D), as there could be dependencies across same-subject electrodes. How was the classifier trained? When running multiple tests, did the authors account for multiple comparisons?

*Reviewer #2:*

Chandrasekaran and Nanivadekar et al. present the first clinical results demonstrating the efficacy of epidural spinal cord stimulation to restore targeted somatosensation in amputees. While other groups are pursuing similar goals using technologies such as intraneural stim, the authors' approach has a lot of promise, in my opinion, because it is minimally invasive and employs clinically-available technology with well-established surgical protocols.

I think this is a nice study that highlights both the promise and challenges of pushing this technology towards clinical adoption. It works well as a proof of concept to show that this minimally-invasive technology can be leveraged for sensation, and that. With only four subjects, and with considerable inter-subject differences, the authors are limited in their ability to show how reliably they can target specific sensations. However, it points the way towards a wide range of follow-up studies to continue to explore the potential of this technology.

1) Overall, I appreciated the depth of the data presented, though occasionally the text could read like a "data dump". While this approach is preferable to the overselling that has come to dominate scientific literature, of course, there were some times where I was not sure why the analysis or experiment was performed and how it helped the case for applying epidural stimulation to amputee patients. This minor criticism could be addressed with a small number of additional clarifications to better guide the reader towards the authors' interpretation of the results (I suggest a couple of specific places below).

2) The authors could make better use of the available figures. I certainly don't advocate that a paper must unnecessarily swell to fill the limits on figures, but there were times where interesting analyses were merely described in text with no visually presented data (for example, the GLM model showing the effect of stimulation amplitude), or where supplementary figures were referenced with no corresponding main figure component (for example the linear relationship between centroid stability and time).

3) Lastly, were there ever motor consequences from the stimulation? The work by Capogrosso and colleagues highlight how epidural activation of sensory afferents can be used to drive movements. This level of interaction could complicate the design of protocols meant to simply restore sensation.

*Reviewer #3:*

"Sensory restoration by epidural stimulation of dorsal spinal cord in upper-limb amputees" is about the sensory feedback restoration to amputees through the epidural Spinal Cord stimulation. Authors describe a human testing with four volunteers is presented in the study.

It is a reasonable study, within the field of sensory substitution/restoration. Yet some aspects should be clarified.

– Regarding the modulation of the perceived sensation: "Increasing the stimulus amplitude increased the perceived intensity of the sensation in all subjects." "For every unit increase in the normalized amplitude, there was a 16% increase in area (p<0.01) and a 110% increase in intensity (p<0.01) across all subjects. This indicates that while percept area is not entirely independent of stimulation amplitude, the unit change in intensity is almost an order of magnitude larger than the unit change in area with respect to stimulation amplitude."

This is somehow misleading, and should be placed in the functional context: it means that in order to modulate the VAS sensation from 1 to 2 the area is increasing the 16% (and what is it on the hand representation spatially)? And therefore, to arrive to the sensation level 2 the area would be 32% bigger? If we consider Figure 2. And subject 2 or 3, it means that the sensory percept already almost whole hand, would then go over the resting part of it? In conclusion it means that it is possible to modulate rather "whole hand" sensation only?

– It is very important to understand the types and percentages of the evoked percepts. Supplementary Figure 3 is meant to do this: why authors do not use some scientific and meaningful representation? E.g. pie chart or something similar with representative percentages? Present figure is disabling the full understanding of sensations variety evoked and their numbers.

– Supplementary Figure 4 is potentially very important since dealing with the stability. Yet the way of presentation of A and B) is really unclear, and not helping in the interpretation of the results. "B, each point represents the change in area of the evoked percept when compared to the median area for a given electrode, expressed as a fraction of the total area of the hand."

Figures should be helpful and easy to interpret, but here I struggle to understand even the meaning of it. It would be important to present it in some more intuitive fashion.

– " suggesting the approach is amenable to a diverse population of amputees."

this is clearly an overstatement: any except very proximal amputees would use other available approaches that deliver more stable, selective, natural and repeatable sensations as extensively reported. Please moderate the statements, from Abstract and within the manuscript.

– Subject 4 is very different w.r.t. other 3-what is the reason? Different surgery?

– Were any types of placebo-s executed? For instance under-threshold stimulation, or any type of possible falsifying strategies (e.g. short pulse supra-threshold and then longer part of pulse under threshold)?

– Clinically, why do you believe that high amputees, e.g. shoulder disarticulation (which are clearly the unique targetable category) would prefer this w.r.t. targeted reinnervation for instance?

– Authors state: "In Subjects 2 and 3, most percepts were accompanied by a sensation on the residual limb… At threshold, paired sensations (perceived in the hand and residual limb) occurred in 0%, 92%, 98% and 8% of all reported sensations for Subjects 1-4 respectively."

So it means that in 2 users the vast majority of all sensations elicited where always accompanied by (at least) a second referred sensation Such a situation can clearly affect the eventual usability of these sensations for the reliable bidirectional control. What is the idea of authors to overcame this?

---

## [Author Response]

Reviewer #1:While there was a lot to like in this study, there were also some major limitations that need to be considered. First, the spatial stability of the stimuli over weeks was pretty disappointing. Is it true that this is comparable with the current state of the art for peripheral stimulation techniques? Second, as emphasised in the manuscript, the inter-individual differences are massive. Because the study only involves 4 subjects, it’s hard to determine what is driving the differences between the subjects, which might be very important for considering the translational power of this technique (e.g. specificity of stimulation site). Third, while this approach is designed for improving prosthesis usage, the current design/results do not seem to take into consideration the practical considerations of prosthesis usage (e.g. active movements, spatiotemporal profile of tactile feedback). For all these reasons, based on the evidence presented here, I'm not convinced that this approach is entirely feasible/promising for tactile restoration of touch. This is not to say that the manuscript should not be published – this is conceptually a very appealing idea and it’s important to set the record clear on the advantages and limitations that it entails. But the manuscript, and Abstract and Discussion in particular, should be changed to reflect these limitations better, to produce a more balanced perspective on the presented results.

Thank you for these thorough and helpful comments.

With regards to spatial stability, we have reworked Figure 6 (now Figure 7) to include summary panels for threshold stability, and the change in area and centroid location between consecutive weeks to better convey that the evoked sensations are stable over the four weeks of the study. We have also updated text in the Results section to describe these new figure panels.

Results:

“We assessed the stability of each evoked percept throughout the duration of the study (e.g. Figure 7A) in terms of the threshold charge (Figure 7B) for evoking a percept in the missing hand. […] For centroid migration, there was a decrease during weeks 2 (β = -23.224, p < 0.05) and 3 (β = -40.585, p < 0.01).”

With regards to intersubject variability, we agree that this is an important point to consider moving forward, and we believe that our manuscript does not identify a clear set of principles for targeting specific regions of the hand and arm with SCS, but that we demonstrate that the techniques show promise and require future development (e.g. of new electrodes with smaller contacts and tighter spacing) to fully characterize the potential for translation of SCS. We have added language in the Discussion to clearly describe the limitations that were identified in this study as well as a path for future development :

Discussion:

“As compared to other techniques that focus on peripheral nerve stimulation, such as epineural stimulation with cuff electrodes or penetrating stimulation with Utah arrays or longitudinal intrafascicular electrodes, our results demonstrate substantially less focal percepts and less consistent coverage of each individual digit across subjects. […] Future work will focus on computational studies to explore this question and design new electrodes that can more selectively target the sensory afferents in the dorsal rootlets to maximize the selectivity and focality of our approach.”

Towards the reviewer’s point about the practical considerations of prosthesis usage, we agree with this limitation, although we expect that, because we are likely activating the central projections of the same primary afferent neurons as occurs with peripheral nerve stimulation, concerns about the spatiotemporal dynamics of stimulation should be similar. We have added panel F to Figure 4 (formerly Figure 3) to demonstrate that our subjects should have similar sensory discrimination capability to those reported by Raspopovic, et al., on an intensity discrimination task, which suggests they may also achieve similar performance with sensory feedback. We have updated the Results, Discussion, and Materials and methods sections to reflect these points.

Results:

“We constructed a confusion matrix to estimate how reliably subjects can distinguish sensations for each of these amplitude ranges. Figure 4F shows the confusion matrix for a 3-target task where the overall accuracy was 72%. […] In the context of clinical translation, these results indicate that it may be possible to provide closed-loop sensory feedback via SCS by mapping the discretized stimulation amplitude to three specific targets.”

Discussion:

“We also observed that subject responses could accurately be separated into three, but not five, separate intensity categories (i.e. low, medium, and high) based on the stimulation amplitude, which suggests that they would be able to successfully perform a 3-level discrimination task, such as identifying three different object stiffness levels. These expected performance levels are similar to those demonstrated by others using peripheral nerve stimulation to restore somatosensation (Raspopovic et al., 2014), and it is possible that they could improve with time and training. Future work should focus on demonstration of such closed-loop control with sensory feedback using lateral SCS.”

Materials and methods:

“We performed a *post-hoc* analysis to determine the maximum number of intensities a subject would likely be able to discriminate, we binned amplitude and magnitude estimation data into multiple bins. […] The normalized confusion matrices for all electrodes were summed to estimate how reliably subjects would be able to distinguish feedback at three or five different amplitudes (Figures 4F, 4—figure supplement 2).”

1) Beyond the translational significance of this manuscript, it offer modest innovation for basic neuroscience (e.g. the observation on the somatotopy across dermatomes). So it might suit better the methodological format Tools and resources? Note that this will require the authors to share some of their code (e.g. for capturing the sensory percepts).

We appreciate this suggestions and have converted the manuscript to the Tools and Resources format and are sharing the code used for capturing sensory percepts. This interface can be downloaded from https://github.com/pitt-rnel/perceptmapper. We have updated the text in the Materials and methods to highlight this.

Materials and methods:

“This interface can be downloaded from https://github.com/pitt-rnel/perceptmapper.”

2) Practically, for the purpose of tactile feedback for prosthesis usage, is this procedure producing sufficient spatial resolution? In other words, are the percepts focal enough, and spatially synchronised sufficiently? Are these percept impacted when the subject is mobile?

While we do not present definitive data here to demonstrate that the sensations evoked by SCS have sufficient spatial resolution to restore tactile feedback, they are similar in spatial resolution to those reported by Marasco et al., 2009, and Schofield et al., 2020, with targeted sensory reinnervation, which has recently been demonstrated as an effective means to provide sensory feedback for a bidirectional prosthetic limb. While we have not carefully characterized the effects of movement on the evoked percepts, we also did not control posture during perceptual or psychophysical tasks, and the consistency of the reported sensations suggests that they are robust to moderate movement, although this is certainly an important area of future research. We have updated the discussion to emphasize these points.

Discussion:

“Importantly though, they are comparable in focality to those used to effectively deliver sensory feedback during recent long-term take-home studies of bidirectional prosthesis using peripheral nerve stimulation (Cuberovic et al., 2019) or targeted reinnervation (Schofield et al., 2020).”

Discussion:

“An important limitation of this study is that we focused on characterizing the sensations evoked by SCS but did not demonstrate that those sensations could be used as part of a closed-loop neuroprosthetic system. While we demonstrate that many of the qualities of the evoked sensations are similar to those reported by others (e.g. sensation intensity modulates linearly with stimulation amplitude), it will be critical to demonstrate that sensations remain stable and are useful during closed-loop prosthetic applications. For example, while we did not control subject posture during any of our experiments, it will be important to demonstrate that sensations remain stable during intentional movements of the neck, shoulders, and arms. Certainly, future work will focus on achieving these demonstrations and characterizing the effects of sensory restoration via SCS on dexterous control of prosthetic limbs.”

3) The authors show very impressive performance with the bayesian classifier for predicting the categorical description, but could this be put into better spatio-temporal context, relating to the other key attributes of the percept? Also, could they correctly classify the sensations across sessions/weeks?

All reviewers had substantial questions about the GLM analysis, including a suggestion from reviewer 2 that we include additional visualizations of the results of the analysis. Upon revisiting these results, we determined that they were likely heavily biased by inter-subject variability. As can be seen in the new Figure 3, each subject reported a dominant set of descriptors, and these dominant sets were highly variable across subjects, which likely contributed the strong performance of the GLM. Because of this issue, we have decided to remove the GLM analysis from the manuscript, and replace it with the sunburst plots (Figure 3) suggested by reviewer 3. This visualization makes clear the relative rates of paresthetic, mixed, and naturalistic sensations in each subject. We have also edited the text to clarify that, within the parameter ranges we tested, there was not a clear relationship across subjects between stimulation parameters and sensation naturalness.

Further, with regards to the relationship between amplitude, intensity, and area of evoked sensations, we replaced the GLM with a set of linear regression-based analyses to compare the relative slopes of the intensity vs. amplitude and area vs. amplitude relationships, and now report these results in Figure 5B which shows that, for most electrodes, amplitude has no effect on area but modulates intensity.

In addition to deleting descriptions of the GLM, we have substantially edited the following text:

Results:

“We asked the subjects to describe the evoked sensations using a set of words provided in a predefined list (Table 2). […] Subject 1 never reported these naturalistic sensations, which could be because we never stimulated at frequencies below 20 Hz, while Subjects 2 and 4 reported naturalistic, mixed, and paresthetic sensations independent of the stimulus frequency.”

Results:

“Figure 5A shows an example of a percept where the area and centroid remain stable as the stimulation amplitude is increased. […] This indicates that for most electrodes, it is possible to modulate percept intensity independent of percept area.”

4) The statistical analysis requires further details. What were the parameters used for the GLM? How were they modelled (e.g. repeated model, fixed, etc)? authors should be weary of collapsing data across electrodes and participants (fixed effect, e.g. Figure 3D), as there could be dependencies across same-subject electrodes. How was the classifier trained? When running multiple tests, did the authors account for multiple comparisons?

We appreciate this comment, and as described above, have removed the GLM analysis from the paper. Elsewhere in the paper (e.g. fitting psychometric curves; see below) we have added additional detail about statistical results.

Reviewer #2:1) Overall, I appreciated the depth of the data presented, though occasionally the text could read like a "data dump". While this approach is preferable to the overselling that has come to dominate scientific literature, of course, there were some times where I was not sure why the analysis or experiment was performed and how it helped the case for applying epidural stimulation to amputee patients. This minor criticism could be addressed with a small number of additional clarifications to better guide the reader towards the authors' interpretation of the results (I suggest a couple of specific places below).

We appreciate this encouraging feedback and have made changes throughout the Results and Discussion to improve clarity and as suggested below. These changes include the following:

Results:

“We sought to determine if stimulation of specific regions of the spinal cord consistently evoked sensations that were perceived to emanate from specific regions of the arm and hard across subjects. We hypothesized that the location of the perceived sensation would be driven by the location of the cathodic electrode with respect to the spinal cord according to expected dermatomes.”

Results:

“This indicates that for most electrodes, it is possible to modulate percept intensity independent of percept area.”

2) The authors could make better use of the available figures. I certainly don't advocate that a paper must unnecessarily swell to fill the limits on figures, but there were times where interesting analyses were merely described in text with no visually presented data (for example, the GLM model showing the effect of stimulation amplitude), or where supplementary figures were referenced with no corresponding main figure component (for example the linear relationship between centroid stability and time).

We appreciate the reviewers comment and have added multiple panels and new figures, including Figure 3 which shows the number of sensations with each modality descriptor for each subject, Figure 4F which shows the accuracy with which we could separate subjects’ reports of intensity into three categories (low, medium, and high) based on the amplitude of stimulation, Figure 5 to visualize the relationship between stimulation amplitude and percept intensity across all electrodes, and Figure 7 B-D, which provide summary visualizations of sensation stability data across weeks.

As described in our response to reviewer 1, we have also removed the GLM analyses from the manuscript. Per your suggestion, we created visualizations of the GLM results, and upon revisiting these results, we determined that they were likely heavily biased by inter-subject variability. As can be seen in the new Figure 3, each subject reported a dominant set of descriptors, and these dominant sets were highly variable across subjects, which likely contributed the strong performance of the GLM. Because of this issue, we have decided to remove the GLM analysis from the manuscript, and replace it with the sunburst plots (Figure 3) suggested by reviewer 3. This visualization makes clear the relative rates of paresthetic, mixed, and naturalistic sensations in each subject. We have also edited the text to clarify that, within the parameter ranges we tested, there was not a clear relationship across subjects between stimulation parameters and sensation naturalness.

Further, with regards to the relationship between amplitude, intensity, and area of evoked sensations, we replaced the GLM with a set of linear regression-based analyses to compare the relative slopes of the intensity vs. amplitude and area vs. amplitude relationships, and now report these results in Figure 5B now presents these results and shows that, for most electrodes, amplitude has no effect on area but modulates intensity.

3) Lastly, were there ever motor consequences from the stimulation? The work by Capogrosso and colleagues highlight how epidural activation of sensory afferents can be used to drive movements. This level of interaction could complicate the design of protocols meant to simply restore sensation.

This is an excellent question and certainly could be a concern if large motor contractions were evoked by epidural SCS. While high amplitude stimulation can evoke strong reflexive contractions, we consistently found that the perceptual threshold was lower than the threshold for reflexive activity and all of the sensations described in this study occurred without any concomitant visible muscle contractions. We did not measure evoked EMG in the residual limb during this study, although this will be important future work. We have added the following text to the Results and Discussion to describe these findings and future work:

Discussion:

“While SCS did not evoke overt reflexive movements of the residual limb in any subject at the stimulation amplitudes used in this study, it is possible that these proprioceptive percepts result from small reflexive contractions of residual limb muscles which themselves activate muscle spindle afferents.”

Reviewer #3:"Sensory restoration by epidural stimulation of dorsal spinal cord in upper-limb amputees" is about the sensory feedback restoration to amputees through the epidural Spinal Cord stimulation. Authors describe a human testing with four volunteers is presented in the study.It is a reasonable study, within the field of sensory substitution/restoration. Yet some aspects should be clarified.

We appreciate this feedback and have made changes throughout the manuscript to reflect these suggestions.

– Regarding the modulation of the perceived sensation: "Increasing the stimulus amplitude increased the perceived intensity of the sensation in all subjects." "For every unit increase in the normalized amplitude, there was a 16% increase in area (p<0.01) and a 110% increase in intensity (p<0.01) across all subjects. This indicates that while percept area is not entirely independent of stimulation amplitude, the unit change in intensity is almost an order of magnitude larger than the unit change in area with respect to stimulation amplitude."This is somehow misleading, and should be placed in the functional context: it means that in order to modulate the VAS sensation from 1 to 2 the area is increasing the 16% (and what is it on the hand representation spatially)? And therefore, to arrive to the sensation level 2 the area would be 32% bigger? If we consider Figure 2. And subject 2 or 3, it means that the sensory percept already almost whole hand, would then go over the resting part of it? In conclusion it means that it is possible to modulate rather "whole hand" sensation only?

This is an excellent point. As mentioned above, we have replaced this GLM analysis with a set of linear regression-based analyses to compare the relative slopes of the intensity vs. amplitude and area vs. amplitude relationships, and now report these results in Figure 5B now presents these results and shows that, for most electrodes, amplitude has no effect on area but modulates intensity.

– It is very important to understand the types and percentages of the evoked percepts. Supplementary Figure 3 is meant to do this: why authors do not use some scientific and meaningful representation? E.g. pie chart or something similar with representative percentages? Present figure is disabling the full understanding of sensations variety evoked and their numbers.

Thank you for this suggestion. We have changed supplementary figure 3 from a word cloud to a sunburst plot and have included it as a Figure 3. We have also updated the text with additional detail about the modality of sensations as follows.

Results:

“All sensations that had an “electrical tingle”, “pins and needles”, “sharp” or “tickle” component were considered paresthetic. […] In contrast, touch-like sensations reported in Subjects 2 and 4 were commonly accompanied by a paresthesia.”

– Supplementary Figure 4 is potentially very important since dealing with the stability. Yet the way of presentation of A and B) is really unclear, and not helping in the interpretation of the results. "B, each point represents the change in area of the evoked percept when compared to the median area for a given electrode, expressed as a fraction of the total area of the hand."Figures should be helpful and easy to interpret, but here I struggle to understand even the meaning of it. It would be important to present it in some more intuitive fashion.

As per the reviewer’s suggestion, we have removed Figure S4 and added three new panels to Figure 7 providing additional detail about the stability of evoked sensations. This includes a panel showing that sensory thresholds were stable across weeks, a panel showing that most centroids migrated by 2 cm or less between weeks and that this migration decreased over time, and a panel showing that the area for most sensations either became smaller or stayed the same between weeks. We have also updated the Results section to reflect these changes.

Results:

“We assessed the stability of each evoked percept throughout the duration of the study (e.g. Figure 7A) in terms of the threshold charge (Figure 7B) for evoking a percept in the missing hand. […]We found a significant decrease in area over time across all weeks, (β = -0.201, p < 0.01). For centroid migration, there was a decrease during weeks 2 (β = -23.224, p < 0.05) and 3 (β = -40.585, p < 0.01).”

– " suggesting the approach is amenable to a diverse population of amputees."this is clearly an overstatement: any except very proximal amputees would use other available approaches that deliver more stable, selective, natural and repeatable sensations as extensively reported. Please moderate the statements, from Abstract and within the manuscript.

As suggested by reviewer 1 and the *eLife* editors, we have converted the manuscript to the Tools and Resources format, which includes highlighting both the relative strengths and weaknesses of our approach with respect to other techniques for sensory restoration (e.g. peripheral nerve stimulation). While we agree that percutaneous spinal cord stimulation has weaknesses with regards to selectivity and possibly with regards to naturalness, as compared to other techniques, we do not believe that other techniques have demonstrated substantially superior stability or repeatability as compared to this technique. Further, we feel that there is a critical advantage in the minimally invasive nature of SCS for translatability. It is important to note, however, that our long-term goal is not to develop percutaneous SCS with commercially available electrodes as a somatosensory neuroprosthesis, but rather to use these techniques as a method for achieving preliminary technical and scientific development towards a future fully implantable system with electrodes that are better suited for our goals, such as highly focal sensations. We have made edits throughout the Abstract, Introduction, and Discussion sections to clarify the goals and impact of the study with respect to these points.

– Subject 4 is very different w.r.t. other 3-what is the reason? Different surgery?

This is a keen observation that merits additional discussion. For Subject 4, monopolar stimulation evoked sensations that were at least as focal as those evoked by multipolar stimulation in the other three subjects. Further, Subject 4 was the only participant that reported consistent proprioceptive percepts on any electrodes. We were not able to identify any particular aspect of the surgical approach or characteristic of the subject that could be associated with the focality of the evoked percepts. It is possible that the trans-radial nature of the amputation allowed for reflexive activation of muscles in the residual limb which indirectly evoked proprioceptive percepts. As we mention in the Discussion, future work should focus on the use of subject-specific computational models to determine the locations of activation on the spinal cord for particular stimulation parameters and associate those with the percepts they evoked to develop a better understanding of the effects of SCS on our neural targets. To highlight this difference and address the reviewer’s questions, we have added the following text to the Discussion section:

Discussion:

“We could not identify any difference (e.g. in surgical technique) that led to this difference in focality of monopolar stimulation for Subject 4 as compared to all other subjects.”

Discussion:

“Only one subject (Subject 4, trans-radial amputation) reported proprioceptive percepts that were repeatedly evoked over more than a few minutes. […]Regardless, we propose that even though we evoked primarily paresthetic sensations, the ability to evoke these percepts via a clinically translatable approach in individuals with high-level amputations establishes the promise of this approach towards restoring sensation.”

– Were any types of placebo-s executed? For instance under-threshold stimulation, or any type of possible falsifying strategies (e.g. short pulse supra-threshold and then longer part of pulse under threshold)?

We agree that it is important to minimize bias in these perceptual experiments and did use randomization techniques throughout the study to minimize bias as much as possible. When recording perceptual responses, we randomized the selection of stimulation electrodes and amplitudes across trials. We have added text below to describe this randomization. For psychophysical experiments (i.e. threshold detection and just-noticeable differences) we randomized the interval that included stimulation or reference amplitude. This randomization is described in the Materials and methods section:

Materials and methods:

“The order of stimulation electrodes and amplitudes was randomized to prevent subjects from predicting the location and perceptual qualities of sensations from previous trials.”

Materials and methods:

“Two one-second-long windows, separated by a variable delay period, were presented and indicated by a change in the color of the fixation cross. Stimulation was randomly assigned to one of the two windows.”

Materials and methods:

“One of the stimulation amplitudes in every trial was held constant while the other was chosen randomly from a list of stimulus amplitudes constituting a block. The constant amplitude was either fixed at 2.5 mA for the lower standard amplitude or at 4.0 mA for the higher standard amplitude. The windows in which standard and the test amplitude were administered was randomized as well. This block of stimulus amplitudes was repeated up to 8 times and the presentation sequence was randomized within each block.”

Materials and methods:

“A block of stimulus amplitudes consisted of 6-10 values linearly spaced between the detection threshold of the electrode being tested and the highest value that did not evoke a painful percept up to 6 mA. This block of chosen amplitudes was presented six times and the presentation sequence was randomized within each block.”

– Clinically, why do you believe that high amputees, e.g. shoulder disarticulation (which are clearly the unique targetable category) would prefer this w.r.t. targeted reinnervation for instance?

We believe that this widespread use of spinal cord stimulation and the relative minimally invasive nature of the approach make these techniques potentially attractive, not just to those with high-level amputations, but to all people with limb amputation. As mentioned above, we have made edits throughout the manuscript to highlight the relative strengths and weaknesses of the approach as compared to peripheral nerve stimulation and targeted reinnervation. We have specifically highlighted the surgical advantages to the approach as follows:

Discussion:

“Another major advantage of the approach is its viability for people with high-level amputations, in which the peripheral nerve has been amputated. In this population, the spinal cord and roots typically remain intact, and we have demonstrated that stimulation of those structures can produce focal sensations in the missing hand. Currently, the only other viable neuroprosthetic techniques for restoring sensation after proximal amputation are invasive approaches such as targeted reinnervation or stimulation of structures in the central nervous system.”

– Authors state: "In Subjects 2 and 3, most percepts were accompanied by a sensation on the residual limb… At threshold, paired sensations (perceived in the hand and residual limb) occurred in 0%, 92%, 98% and 8% of all reported sensations for Subjects 1-4 respectively."So it means that in 2 users the vast majority of all sensations elicited where always accompanied by (at least) a second referred sensation Such a situation can clearly affect the eventual usability of these sensations for the reliable bidirectional control. What is the idea of authors to overcame this?

This is certainly an important point, and it is currently unclear whether this is a limitation of the electrodes used in this study (i.e. because of their relatively large size) or a fundamental limitation of lateral epidural SCS. We have added text in the Discussion to highlight this point as follows:

Discussion:

“As compared to other techniques that focus on peripheral nerve stimulation, such as epineural stimulation with cuff electrodes or penetrating stimulation with Utah arrays or longitudinal intrafascicular electrodes, our results demonstrate substantially less focal percepts and less consistent coverage of each individual digit across subjects. […] Future work will focus on computational studies to explore this question and design new electrodes that can more selectively target the sensory afferents in the dorsal rootlets to maximize the selectivity and focality of our approach.”